# Structure- and computational-aided engineering of an oxidase to produce isoeugenol from a lignin-derived compound

Yiming Guo [1], Laura Alvigini[2], Milos Trajkovic [1], Lur Alonso-Cotchico[3], Emanuele Monza[3], Simone Savino [1], Ivana Marić[1], Andrea Mattevi [2] & Marco W. Fraaije [1] ✉

Various 4-alkylphenols can be easily obtained through reductive catalytic fractionation of lignocellulosic biomass. Selective dehydrogenation of 4-*n*-propylguaiacol results in the formation of isoeugenol, a valuable flavor and fragrance molecule and versatile precursor compound. Here we present the engineering of a bacterial eugenol oxidase to catalyze this reaction. Five mutations, identified from computational predictions, are first introduced to render the enzyme more thermostable. Other mutations are then added and analyzed to enhance chemoselectivity and activity. Structural insight demonstrates that the slow catalytic activity of an otherwise promising enzyme variant is due the formation of a slowly-decaying covalent substrate-flavin cofactor adduct that can be remedied by targeted residue changes. The final engineered variant comprises eight mutations, is thermostable, displays good activity and acts as a highly chemoselective 4-*n*-propylguaiacol oxidase. We lastly use our engineered biocatalyst in an illustrative preparative reaction at gram-scale. Our findings show that a natural enzyme can be redesigned into a tailored biocatalyst capable of valorizing lignin-based monophenols.

Lignin is an abundant and renewable source of phenolic compounds, and represents a sustainable alternative stream to aromatic chemicals currently derived from the petroleum industry[1–3]. However, due to its structural complexity, lignin remains insufficiently exploited for the synthesis of high-value products. Reductive catalytic fractionation (RCF) is an emerging technology which allows efficient production of phenolic monomers from lignocellulosic biomass by lignin extraction, depolymerization, and stabilization[4]. RCF yields a mixture of lignin-derived phenols. By tuning process conditions and depending on the type of lignocellulose starting material, a variety of para-substituted phenols can be obtained, often with 4-*n*-propylguaiacol as a major product[5,6]. As this phenol, to our knowledge, is not used in any industrial-scale (bio)chemical process, it is attractive to develop new chemical routes that exploit this abundant lignin-derived compound as starting material[1]. Notably, 4-*n*-propylguaiacol is highly similar to

isoeugenol, only lacking an alkene double bond in the propyl chain. Isoeugenol is known as a plant-derived flavor and fragrance. For instance, this compound is widely used in deodorants[7]. Furthermore, isoeugenol can also be used for the synthesis of other valuable compounds such as vanillin[8,9], fine chemicals[10,11], polymers[12], and epoxy resins[13]. A one-step selective dehydrogenation of 4-*n*-propylguaiacol into isoeugenol would enable the lignin-based production of this valued monophenol (Fig. 1)[11,14,15].

Many enzymes acting on aromatic compounds have been discovered and identified as a result of studies on the fungal and microbial catabolism of naturally occurring aromatic monomers and polymers[16,17]. Only a subset of these enzymes is capable of converting alkylphenols. There is one well-studied group of flavin-dependent oxidative enzymes acting on 4-alkylphenols, named after vanillyl alcohol oxidase (VAO). The VAO subfamily includes vanillyl alcohol

[1]Molecular Enzymology Group, University of Groningen, Groningen, the Netherlands. [2]Department of Biology and Biotechnology "Lazzaro Spallanzani", University of Pavia, Pavia, Italy. [3]Zymvol Biomodeling S.L., Barcelona, Spain. ✉e-mail: m.w.fraaije@rug.nl

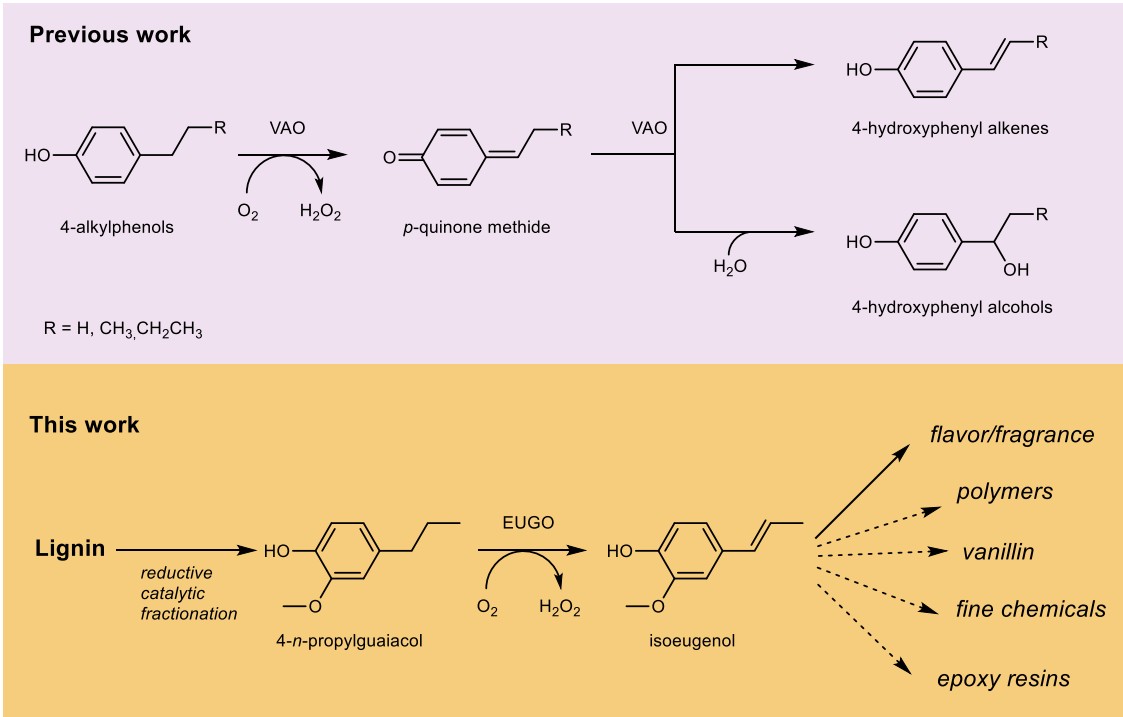

**Fig. 1 | Conversion of 4-alkylphenols to 4-hydroxyphenyl alkenes and 4-hydroxyphenyl alcohols by vanillyl alcohol oxidase (VAO) and lignin-based synthesis of isoeugenol as fragrance, flavor or precursor of other** **high-value products.** The upper panel shows the mechanism by which VAO oxidizes 4-alkylphenols. The lower panel shows the catalytic pathway, which includes an engineered EUGO, by which lignin can be converted into valuable products.

oxidase (VAO), eugenol oxidase (EUGO), 4-ethylphenol oxidase (EPO), eugenol hydroxylase (EH) and *p*-cresol methylhydroxylase (PCMH)[18]. Although these proteins are sequence- and structure-related and all contain a covalently bound FAD cofactor for catalysis, they vary in the type of reactions that they catalyze. Depending on the enzyme, they catalyze alcohol oxidation, hydroxylation, amine oxidation, oxidative demethylation, and dehydrogenation reactions. The VAO-type hydroxylases (PCMH and EH) are not convenient for use as biocatalysts, as they rely on cofactor regeneration and depend on a cytochrome subunit for activity, making it difficult to produce them as recombinant proteins[19]. The VAO-type oxidases (VAO, EUGO and EPO) are instead more attractive as they are composed of only one protomer and require only dioxygen as a co-substrate.

The prototype VAO from the fungus *Penicillium simplicissimum* has been extensively studied[20,21] in the last three decades. When acting on 4-alkylphenols, VAO typically produces a mixture of the corresponding 1-(4'-hydroxyphenyl)alcohols and 1-(4'-hydroxyphenyl) alkenes (Fig. 1)[22,23]. In-depth mechanistic and structural studies have provided a great deal of insight into the active site and the catalytic mechanism. The oxidation of 4-alkylphenols starts with a hydride transfer from the Cα atom of the substrate to the N5 of the flavin cofactor. This is facilitated by deprotonation of the phenolic moiety in the active site by two catalytic tyrosines. Consequently, a *p*-quinone methide is formed and stabilized in the active site. This reactive intermediate can subsequently be hydroxylated by the base-catalyzed attack of water, or tautomerise to the dehydrogenation product. The outcome of the reaction is determined by the relative rates of these competing reactions within the active site. Regio- and stereospecific hydroxylation by water addition is tuned by the water accessibility of the enzyme active site and the orientation of the alkyl side chain of the substrate[24].

For the envisaged reaction of this study, converting 4-*n*-propylguaiacol into isoeugenol (Fig. 1), VAO was not the most obvious candidate. Previous studies showed that this fungal oxidase has a strong preference to hydroxylate instead of dehydrogenate 4-alkylphenols. In addition, its expression in *Escherichia coli* is inferior compared to the expression of bacterial homologs[25,26]. Also, VAO is an octameric enzyme[27] that makes computational analyses quite demanding compared with mono- or dimeric homologs. As an alternative, we focused on the more recently identified and highly homologous bacterial VAO-type oxidase: EUGO from *Rhodococcus jostii* RHA1[26]. Similar to VAO, EUGO is active on various 4-substituted phenols and contains a covalently bound FAD. In contrast to VAO, it is exceptionally well expressed in *E. coli*[28] as homodimeric enzyme. Also, for EUGO, several crystal structures are available to guide enzyme engineering. Another attractive feature is the tolerance of EUGO towards solvents[29]. It has been shown that the enzyme tolerates up to 10% v/v dimethyl sulfoxide (DMSO). Such a characteristic is highly desired when working with poorly soluble compounds in water, such as 4-*n*-propylguaiacol. However, EUGO exhibits weak activity on 4-alkylphenols. This is demonstrated by the relatively low $k_{cat}$ for 4-ethylphenol ($0.0026\,s^{-1}$) as compared with other EUGO substrates ($k_{cat}$ values for eugenol and vanillyl alcohol are $3.1\,s^{-1}$ and $12\,s^{-1}$, resp.). In fact, EUGO was found to show very poor activity toward 4-*n*-propylguaiacol ($k_{cat} = 0.008\,s^{-1}$, vide infra).

In this work, we engineer EUGO, a VAO-type oxidase, to catalyze the selective dehydrogenation of 4-*n*-propylguaiacol to isoeugenol. For this enzyme engineering challenge, we exploit known and newly determined crystal structures and use computational tools and detailed mechanistic insights. In a multi-step enzyme engineering project, (i) the thermostability is improved, (ii) the required selective dehydrogenation activity is introduced, and (iii) structure-based mutations prevent an undesired side reaction (formation of covalent cofactor adduct). The final oxidase variant, efficient in converting 4-*n*-propylguaiacol into isoeugenol, is named 4-*n*-propylguaiacol oxidase (PROGO) and contains eight mutations when compared with the parental enzyme. Structural elucidation of several intermediate variants provides a molecular explanation of the observed effects of the

stepwise introduced mutations. Bioconversion experiments demonstrate that PROGO can be used as a stable biocatalyst to produce isoeugenol as the main product (97% chemoselectivity), which will enable the development of lignin-based production of this high-value phenol.

## Results and discussion

### Engineering a thermostable EUGO variant by computational predictions

Before tuning the activity of EUGO towards 4-$n$-propylguaiacol, we first decided to generate a thermostable EUGO variant. Such variants would be better suited for subsequent enzyme redesign, tolerating destabilizing mutations. For increasing its thermostability, we applied a computational method that we have developed and shown to be effective for various proteins[30–32]; Framework for Rapid Enzyme Stabilization by Computational libraries (FRESCO)[33,34]. Using FRESCO, mutations that render a protein more thermostable are predicted based on computational analyses (including FoldX and Rosetta ddg calculations and molecular dynamic simulations). The respective mutations are experimentally tested, and can be combined in highly stable enzyme variants. In the above procedure, residues located close to the FAD are excluded, to prevent inactivating mutations due to ineffective cofactor binding and/or altering the active site. Amongst the four crystal structures of wild-type EUGO deposited in the protein database bank (PDB: 5FXD, 5FXE, 5FXF, and 5FXP), the structure of EUGO bound to isoeugenol was used (PDB: 5FXD) because it displayed the highest resolution (1.7 Å) and had a relevant ligand bound. Computational analyses were done for a total of 9196 single mutants of EUGO (mutating 484 residues out of the 526 EUGO residues). Of these mutants, 496 passed the energy threshold (calculated $\Delta\Delta G$ lower than −5 kJ mol$^{-1}$). These were then submitted to short molecular dynamics (MD) simulations and a subsequent visual inspection resulted in a selection of 72 predicted stabilizing mutations for experimental testing (Supplementary Table 1).

The library of 72 FRESCO-predicted EUGO single mutants was prepared and subsequently screened for thermostability. Transformed cells were grown in 96-well format and used for purifying all EUGO variants by affinity chromatography. Next, the thermostability of the purified EUGO variants was investigated by thermal unfolding. Thermofluor was employed to measure the melting temperature ($T_m$) of the mutants (Fig. 2). Two mutants (S81D and H174Q) could not be analyzed due to too low expression. Of the remaining 70 mutants, 45 mutants had a similar $T_m$ to wild-type EUGO (−1 °C ≤ $\Delta T_m$ ≤ +1 °C), 16 were destabilizing ($\Delta T_m <$ −1 °C) and 9 were stabilizing ($\Delta T_m >$ +1 °C). Notably, 7 mutations showed a significant improvement ($T_m \geq$ +2 °C). Two thermostable mutants (H434W and H434Y) featured a mutation of the same residue. It was decided to exclude the H434W mutant for further studies as the H434Y mutant displayed a higher thermostability. Although all residues within 5 Å from the FAD were excluded from the FRESCO predictions, distal mutations can still affect activity through distant and indirect interactions. Vanillyl alcohol was used as a test substrate to determine the relative activity of the 6 mutants that displayed a relatively high thermostability (Supplementary Table 2). This revealed that the N202D mutation resulted in a fivefold lower $k_{cat}$ value, while the other five mutants showed similar $k_{cat}$ values compared with wild-type EUGO. Therefore, the N202D mutation was discarded from further studies. The five activity-retaining mutations were then iteratively combined to assess whether their combinations would result in an additive effect on the thermostability of EUGO (Supplementary Table 2). Gratifyingly, each added mutation resulted in a higher thermostability of the resulting EUGO mutant. The resulting fivefold EUGO mutant (EUGO5X) showed an increase of 13.5 °C in thermal stability. EUGO5X retained oxidase activity for vanillyl alcohol, displaying a slightly lower $k_{cat}$ and a fourfold higher $K_M$ (Supplementary Table 2). This robust EUGO variant was used as a template for further enzyme engineering.

### Smart library for chemoselectivity of EUGO on 4-$n$-propylguaiacol

The next challenge was to redesign the active site of EUGO so that it accepts 4-$n$-propylguaiacol as substrate. 4-$n$-Propylguaiacol was docked into the EUGO structure (PDB:5FXD, chain B) using Autodock VINA[35] and the best scoring binding mode was used as starting point for a Rosetta Coupled Moves experiment[36] to identify active site variants with enhanced protein–ligand pre-catalytic interactions. All the positions within 5 Å of the 4-$n$-propylguaiacol and whose side chain was pointing towards the substrate-binding cavity (excluding the catalytic residues) were submitted to in silico mutagenesis individually (Fig. 3a) and evaluated by protein–ligand docking (see the Methods section for further details). For every position, only the amino acid changes present in a multiple sequence alignment in a higher population than 2% were allowed to avoid perturbing the expression and/or folding of the protein. This resulted in a list of 16 suggested single-site mutants that were introduced into both EUGO and EUGO5X: V166A, V166T, Y168A, V276T, M282G, M282S, L381I, L381Q, I391A, I391V, G392A, S394A, S394V, A423L, Q425L, I427V (see Fig. 3b).

Conversions of 4-$n$-propylguaiacol were initially performed using the mutant-containing cell extracts, and assayed by high-performance liquid chromatography (HPLC; Fig. 4a). We monitored the depletion of the substrate as well as the formation of all three possible products; the desired dehydrogenation product ($E$-isoeugenol), the unwanted hydroxylated alcohol (4-(1-hydroxypropyl)−2-methoxyphenol)), and the double-oxidation ketone product formed upon its subsequent oxidation (1-(4-hydroxy-3-methoxyphenyl)−1-propanone)). We first observed that the wild-type EUGO converted 45% of the substrate (5 mM) in 24 h with about 80% of the product being isoeugenol and 20% being a mixture of the undesired alcohol and ketone. Using the wild-type as a benchmark, we found that almost all mutations cause a significant decrease in the biocatalytic performance. However, the two mutations targeting S394 featured very high chemoselectivity for dehydrogenation, with isoeugenol as the dominant (S394A) or only (S394V) detected product. Moreover, the S394A EUGO exhibited relatively high conversions.

Based on these encouraging data, S394A, S394V and four other S394 mutations (S394G/T/P/N) were introduced to the thermostable EUGO5X. The properties of this small collection of selected mutants were analyzed using the purified proteins (Fig. 4b). As first observation, we noticed a clear beneficial effect of stability on conversion: within 3 h, about 75% of starting material was converted by EUGO5X, which compares very favorably to 25% conversion afforded by wild-type EUGO. Yet, a large part (about 33%) of the product was a mixture of undesired alcohol and ketone products. In contrast, the product formed by the S394V-EUGO5X mutant was mainly isoeugenol (>95%) with a conversion of 80%. Also, the other S394 mutants affected the conversion and selectivity, but S394V was superior.

To shed light on the effects of the introduced mutations on the enzyme structure, the crystal structure of the S394V-EUGO5X mutant in complex with 4-$n$-propylguaiacol was solved at 2.9 Å resolution. The asymmetric unit comprises eight enzyme subunits forming four compact dimers. The overall three-dimensional structure of the mutant is virtually identical to that of the wild protein as revealed by an RMSD of 0.49 Å for 526 Cα atoms. This indicates that the six mutations do not result in any significant structural change at the level of the general backbone conformation and, most importantly, in the catalytic core (Supplementary Fig. 1). The improved stability of S394V-EUGO5X compared to the wild-type enzyme is due to the five mutations already present in the EUGO5X mutant (Supplementary Fig. 1, red spheres). The pattern emerging upon investigating these stabilizing mutations is clear: four of them (S81H, A423M, H434Y, S518P) create a better side chain packing by presenting bulkier or rigid aliphatic groups, whilst the fifth mutation (I445D) substitutes a hydrophobic residue with a

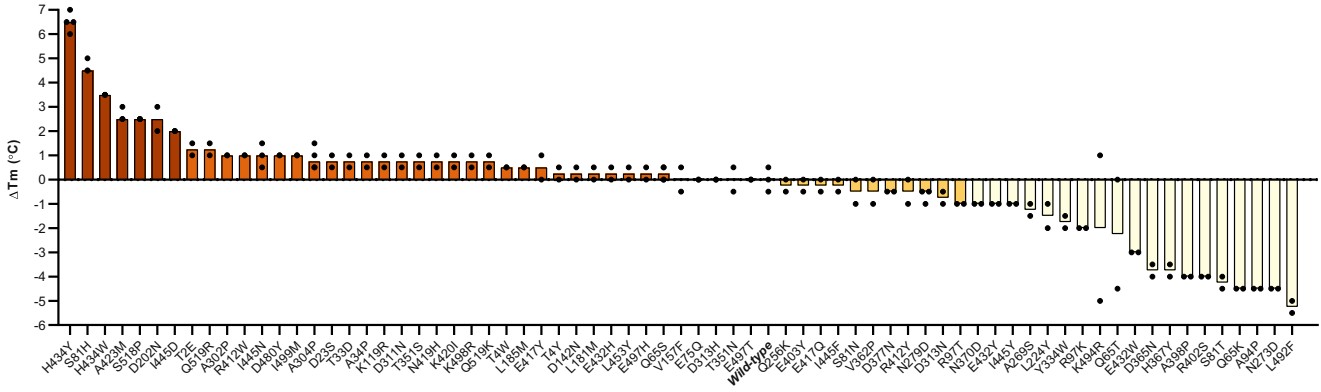

**Fig. 2 | The difference in $T_m$ values for FRESCO-predicted EUGO mutants.** The $T_m$ values of 70 single mutants (2–4 replicates) determined by using the Thermofluor assay are shown with dots and the average $T$m values are shown in bars. The average $T_m$ of wild-type EUGO is 66.5 °C and set as a reference. The seven most stabilizing mutations with a $T_m$ increase of at least 2 °C that were studied in more detail are in dark orange bars. Two of the 72 FRESCO-predicted mutants could not be expressed and were not further studied.

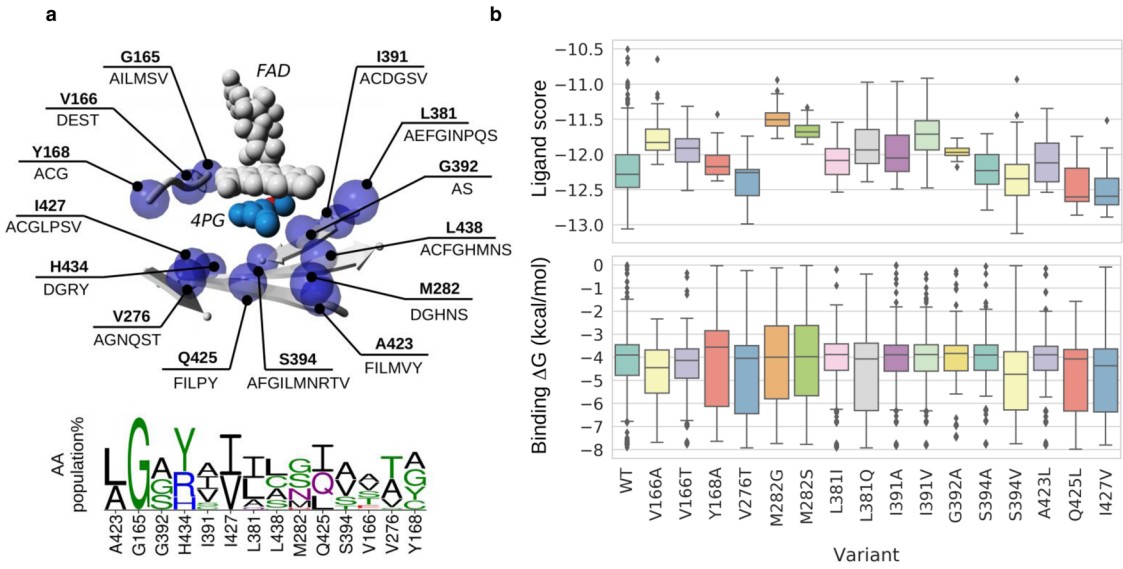

**Fig. 3 | Computational analysis for improving activity. a** The positions at the EUGO active site selected for mutagenesis (blue spheres) and the changes observed in every position according to a multiple sequence alignment (top). The FAD cofactor and the 4-*n*-propylguaiacol (4PG) ligand are represented as gray and light blue balls, respectively. Residues selected by the Monte Carlo algorithm for every position in the Rosetta Coupled Moves experiment are indicated (bottom).

**b** Ligand Scores from Rosetta Coupled Moves (top) and binding energies from Autodock VINA (bottom) for all the selected variants presented in boxplots showing the median of the data as a central line, the box represents the inter-quartile section, which covers from 25 to 75% of the distribution, and the tips represent minimum and maximum outliers.

charged one on the protein surface, and might contribute to overall stability by preventing protein aggregation and/or exposing a more hydrophilic side chain in direct contact with the solvent. The sixth (S394V) mutation of S394V-EUGO5X promotes enhanced chemoselectivity towards isoeugenol formation by favouring substrate dehydrogenation over hydroxylation. As shown in Fig. 5a, V394 is located at the entrance of the cavity where the substrate binds. Notably, this position represents the connection point between the active site and the wide chamber located at the subunit interface, which may comprise the entry passage of the water molecule which is used by EUGO to hydroxylate the reactive quinone methide intermediate. Therefore, a reasonable explanation for the higher chemoselectivity and reduced hydroxylation featured by the S394V-EUGO5X mutant is that the more hydrophobic and tightly sealed cavity of the S394V enzyme may hinder the accessibility of the Cα atom of the *p*-quinone methide intermediate to water, limiting the formation of the unwanted alcohol side-product.

## Structure- and sequence-guided mutations

The engineered sixfold mutant S394V-EUGO5X showed promising selectivity by producing isoeugenol as the main product. However, when determining its steady-state kinetic parameters for 4-*n*-propylguaiacol, it was found to display a low $k_{cat}$ of 0.028 s$^{-1}$ (Table 1). For comparison, wild-type EUGO displays a 100-fold higher activity with eugenol, a compound very similar to 4-*n*-propylguaiacol. This suggested that there was room for improvement, to engineer a more active variant of the S394V-EUGO5X mutant. Some more detailed biochemical studies were performed to understand the molecular basis for the low activity with 4-*n*-propylguaiacol. First, the flavin absorbance at 442 nm of the S394V-EUGO5X mutant was monitored during the conversion of 4-*n*-propylguaiacol. This revealed that the typical absorbance of the oxidized flavin cofactor decreased by 50% during conversion, suggesting that the cofactor is mainly in a reduced state during catalysis. The observation that upon full substrate conversion the flavin cofactor returned to its fully oxidized state indicated

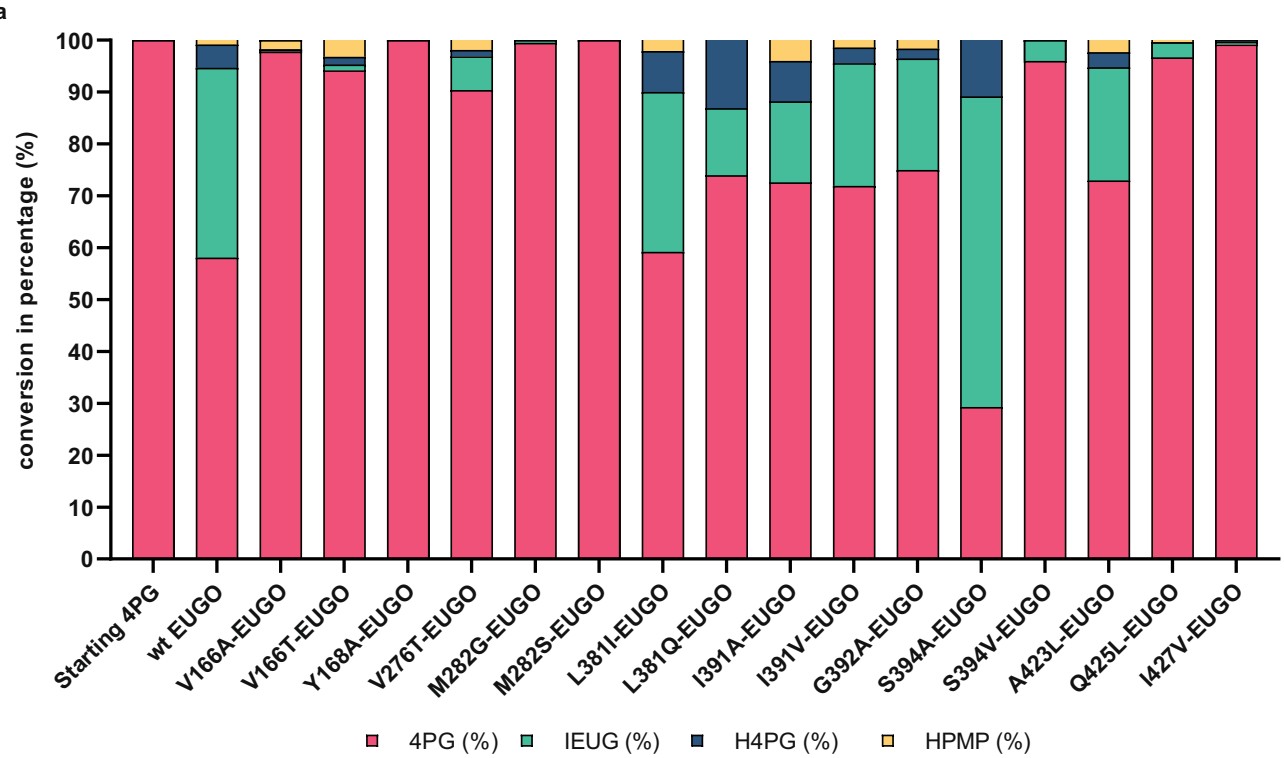

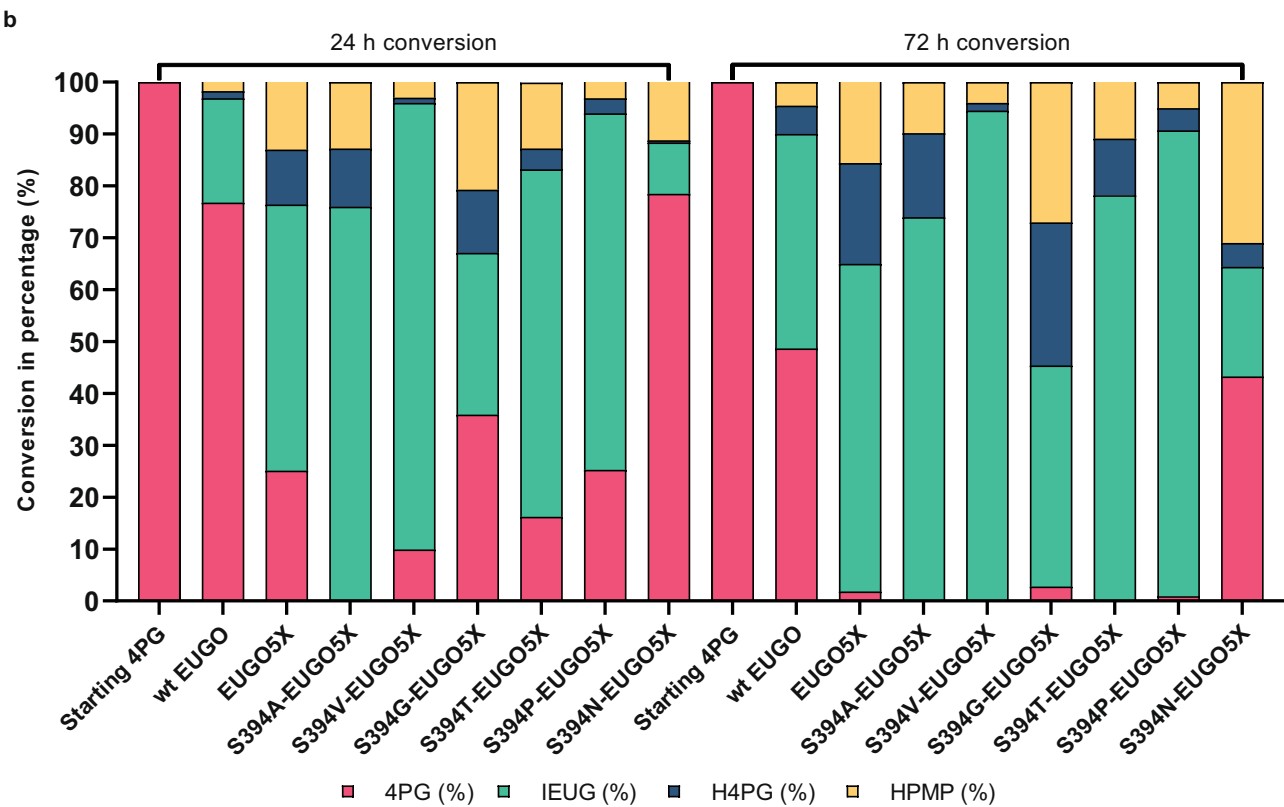

**Fig. 4 | 4-*n*-propylguaiacol conversion by EUGO mutants.** 4-*n*-propylguaiacol (4PG), isoeugenol (IEUG), 4-(1-hydroxypropyl)−2-methoxyphenol (H4PG), 1-(4-hydroxy-3-methoxyphenyl)−1-propanone (HPMP) are shown in red, green, orange and blue, respectively. **a** 24 h 4-*n*-propylguaiacol conversion by cell-free extracts of EUGO single mutants; **b** 24 h and 72 h 4-*n*-propylguaiacol conversion by purified mutants of EUGO5X in order to improve chemoselectivity. Conversions of 4-*n*-propylguaiacol (5 mM) were carried out in the presence of enzyme (10 μM) in a 50 mM KPi buffer containing 10% v/v DMSO at pH 7.5.

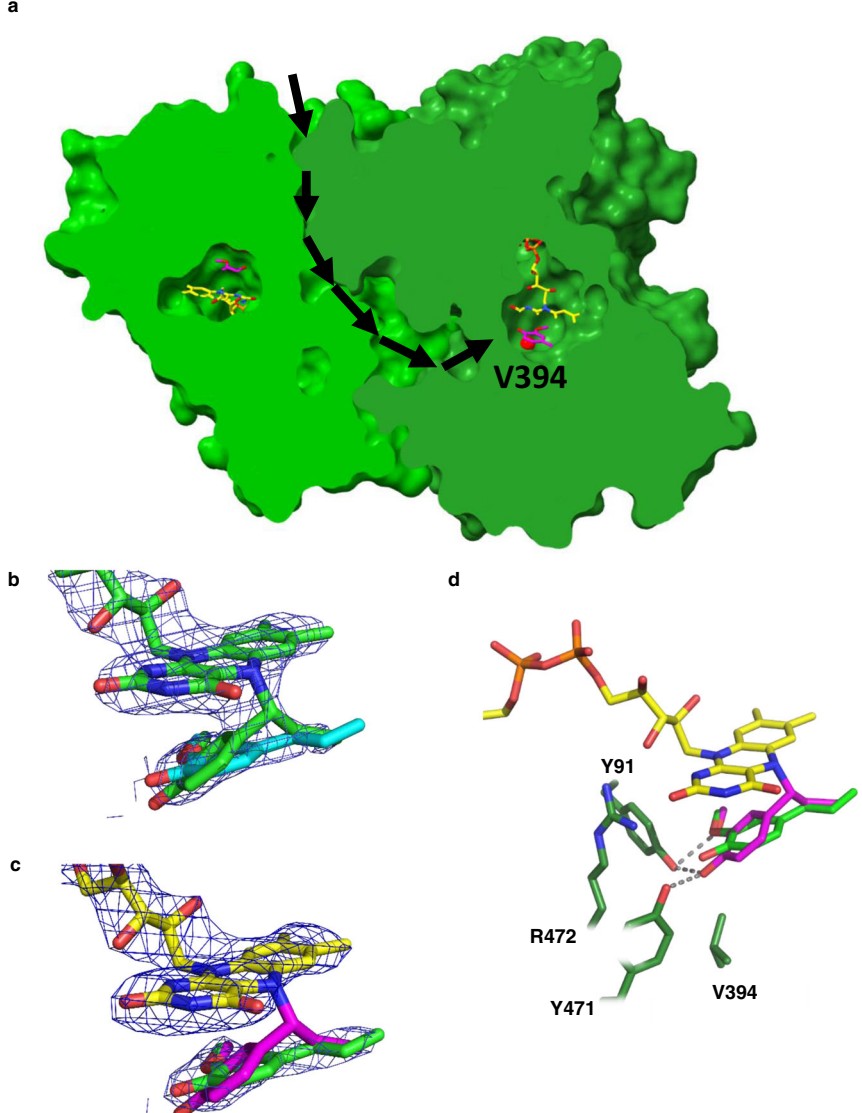

**Fig. 5 | Structural features of S394V-EUGO5X. a** Position of V394 in the cavity, showing a sliced view of S394V-EUGO5X. The mutated V394, shown as a red sphere, is in close contact with a deep passageway buried between the two chains of the dimer. **b, c** Detailed views of the electron densities of 4-*n*-propylguaiacol bound to two S394V-EUGO5X subunits. **b** unbiased 2Fo-Fc electron density map showing the presence of a stretch of electron density extending from the flavin N5 atom and suggesting the presence of a covalent bond between the flavin and 4-*n*-propylguaiacol in subunit B. **c** unbiased 2Fo-Fc electron density map of a non-covalently bound ligand observed in chain A. The maps are contoured at 1.2 σ level and were calculated before the inclusion of the ligands in the refinement. In both panels, the non-covalent and covalent ligands are superimposed. Because of different crystal contacts, the crystalline protein chains might differ with regard to the rates of catalysis, substrate diffusion, or product release explaining why the covalently bound ligand accumulates preferentially in certain protein subunits. **d** Superposition of the covalent (in magenta) and non-covalent (in green) 4-*n*-propylguaiacol complexes as observed in two crystallographically independent subunits.

that the apparent reduced redox state during catalysis is reversible (Supplementary Fig. 2). These features hinted at an undesired enzyme intermediate, as it has been observed for VAO when reacting with 4-methylphenol[34]. VAO was previously found to form a relatively stable covalent 4-methylphenol-flavin N5 adduct, severely limiting the rate of catalysis on the smallest 4-alkylphenol. Indeed, when inspecting the crystal structure of the S394V-EUGO5X mutant bound to 4-*n*-propylguaiacol, a flavin N5 adduct was found in the active site. Specifically, careful inspection of the electron density revealed that the substrate is covalently bound to FAD in two of the eight crystallographic subunits present in the asymmetric unit, whilst in the other chains, the ligand was found to be non-covalently bound (Fig. 5b, c). By comparing the subunits with covalent and non-covalent bound substrates, we found that covalent binding is associated with a 20° tilting of the bound substrate (Fig. 5d). This finding confirmed that the low activity of the S394V-EUGO5X mutant is most likely due to the formation of a slowly decaying covalent adduct in the active site. Such a covalent adduct is also in line with the loss of flavin absorbance observed during catalysis and, importantly, provided a lead to where to focus enzyme engineering in an attempt to destabilize or prevent the formation of this covalent adduct.

For the next step in optimizing the enzyme for activity on 4-*n*-propylguaiacol, we therefore targeted residues close to the N5 of the FAD in an attempt to destabilize or prevent the formation of this covalent adduct. When inspecting the three-dimensional structure of S394V-EUGO5X, two residues stood out: D151 and Q425. Both these residues point to the N5 of the flavin cofactor while coming from opposite sides. Inspired by the variation at position 425 in other sequence-related enzymes, we first created and tested a set of 12 different Q425 mutants (Fig. 6). Most of the mutants exhibited

**Table 1 | Thermostability and kinetic properties of wild-type EUGO and mutants**

| Variants | $T_m$ (°C) | 4-*n*-propylguaiacol | | vanillyl alcohol | |
|---|---|---|---|---|---|
| | | $K_M$ (µM)[a] | $k_{cat}$ (s$^{-1}$)[a] | $K_M$ (µM)[a] | $k_{cat}$ (s$^{-1}$)[a] |
| EUGO | 66.5 ± 0.5 | 3.3 ± 0.9 | 0.008 ± 0.0003 | 43.0 ± 2.2 | 6.8 ± 0.2 |
| S394V-EUGO5X | 78.5 ± 0.0 | 3.4 ± 0.4 | 0.028 ± 0.0003 | 7.1 ± 0.6 | 0.075 ± 0.001 |
| D151E/S394V-EUGO5X | 76.5 ± 0.0 | 3.0 ± 0.2 | 0.080 ± 0.002 | 2.9 ± 0.2 | 0.15 ± 0.002 |
| PROGO | 81.6 ± 0.2 | 7.4 ± 0.4 | 0.43 ± 0.007 | 94.9 ± 7.8 | 2.8 ± 0.07 |

[a]$K_M$ and $k_{cat}$ values are based on rates determined at 7 different substrate concentrations (each in duplicate); for each average, the error was smaller than 5%.

significantly higher conversions when compared with the parent S394V-EUGO5X. However, while several Q425 mutants reached (almost) complete conversion, some chemoselectivity was lost, as they produced about 20% of the unwanted alcohol and ketone products. It is worth noting that the EUGO5X Q425L mutant was present in the small library aimed at improving chemoselectivity (*vide supra*) but was discarded due to a lack of improvement. Yet, the mutation performed better in terms of conversion when it was introduced into the S394V-EUGO5X mutant. This illustrates the power of our computationally guided engineering approach for engineering very specific mechanistic and kinetic facets into otherwise unknowable and complex enzyme active sites. Also, D151 was targeted for mutagenesis as we previously found that the highly chemoselective 4-ethylphenol oxidase from *Gulosibacter chungangensis* features glutamate at this position[37]. The D151E/S394V-EUGO5X mutant was created and analyzed. Interestingly, this additional mutation (D151E) resulted in a similar conversion compared to the parent mutant and similar steady-state kinetic parameters (Table 1), but it displayed exquisite chemoselectivity by producing insignificant amounts of the undesired products (Fig. 6). This prompted us to create the combined mutant D151E/S394V/Q425S-EUGO5X. Gratifyingly, this eightfold EUGO mutant was found to display all desired properties: conversion of 4-*n*-propylguaiacol led to the rapid formation of isoeugenol as the main product (97%). Characterization of the purified enzyme revealed that the $k_{cat}$ had improved by two orders of magnitude relative to wild-type EUGO, whilst retaining the thermostability of EUGO5X mutant (Table 1). The fact that the activity on vanillyl alcohol is still higher may indicate that there is still room for improvement but could also be due to different reactivities of both compounds.

The crystal structure of the D151E/S394V/Q425S-EUGO5X in a complex with 4-*n*-propylguaiacol was solved at 2.4 Å resolution. It showed that the bulkier E151 sidechain points towards the propenyl Cα atom of the substrate, potentially sequestering the reactive quinone methide and obstructing attack of water (Figs. 1 and 7). The Q425S mutation renders the environment on top of the substrate less packed, creating a looser niche around the substrate side chain. This allows the 4-*n*-propylguaiacol to rotate slightly away from the flavin N5 as the substrate side chain can partly occupy the space vacated by removing the Q425 carboxamide group (Fig. 7). As a result, the less constricted and tightly packed environment around the Cα atom of the substrate reduces the formation of the covalent adduct that impairs catalysis. The engineered D151E/S394V/Q425S-EUGO5X mutant was named 4-*n*-propylguaiacol oxidase (PROGO).

**Preparative scale conversion of 4-*n*-propylguaiacol**
To further establish the utility of PROGO as a biocatalyst, we performed conversions on a preparative scale. First, 500 mg (3.0 mmol) substrate was converted using 18.5 mg (0.30 µmol) purified PROGO in a 60 mL reaction volume at 25 °C with 10% dimethyl sulfoxide (v/v) as a cosolvent. HPLC analysis confirmed a nearly complete conversion of starting material into isoeugenol after 48 h (Fig. 8). Only trace amounts of 4-*n*-propylguaiacol were detected. The product was extracted with ethyl acetate and purified by column chromatography resulting in

328 mg (2.0 mmol, 66%) isoeugenol. Next, a conversion of 1.27 g (7.6 mmol) 4-*n*-propylguaiacol was performed using *E. coli* cells expressing PROGO and resuspended in 120 mL (OD$_{600}$ = 29) of 50 mM KPi buffer (pH 7.5) with 10% dimethyl sulfoxide (v/v). Again, a nearly full conversion was obtained after 48 h (Fig. 8), and 524 mg (3.2 mmol, 42%) isoeugenol was isolated by extraction and subsequent column chromatography. The lower isolated yield in whole-cell conversion was probably due to the presence of cell debris which retained part of the product. Optimization of the isolation procedure would be needed for optimal product recovery. These results show that PROGO can be used as an isolated enzyme and in recombinant cells to obtain isoeugenol starting from a lignin-derived phenol.

With the recent developments in RCF of lignin-containing biomass, biocatalysts are needed to produce valuable products from the lignin-derived phenolic molecules. We embarked on an enzyme engineering study with the aim of generating a biocatalyst that can be used for producing a valuable compound, isoeugenol from a product of RCF. The targeted enzyme, EUGO, was used as starting point for several reasons: (1) it can be easily expressed, (2) it does not inactivate due to dissociation of its flavin cofactor, being covalently bound, (3) it requires only oxygen as a co-substrate, and (4) its biochemical features were known in detail, including an established substrate acceptance profile and crystal structures. The targeted conversion required a significant improvement of activity on 4-*n*-propylguaiacol as the wild-type enzyme has an low activity for this phenolic compound. To add to the challenge, the engineered oxidase should be specific in generating only one of the two potential products, the dehydrogenated product, isoeugenol. Thus, hydroxylation activity should be eliminated.

We approached this enzyme engineering challenge by first creating a thermostable variant. Using FRESCO, we successfully engineered a thermostable mutant (5 mutations leading to 15 °C higher $T_m$). The beneficial effects of the FRESCO-predicted mutations could be explained by locally optimized structural features. The fact that the stabilizing mutations are far away from each other also explains why they have an additive effect when combined. This underscores the potential of the computational method. The next round of engineering focused on improving chemoselectivity and was also computationally guided. By screening a small library of active-site mutants, a critical residue (S394) was identified, that tunes the selectivity of the oxidase concerning its ability to perform dehydrogenation or hydroxylation reactions. Inspection of the crystal structure revealed that the additional active site mutation (S394V) alters solvent accessibility of the substrate-binding pocket. The latter may indeed limit the efficiency by which water is able to form the hydroxylated product. When analyzing the sixfold mutant S394V-EUGO5X, it was discovered that it displays a very low activity, which was eventually explained by the formation of an inactivating substrate-flavin covalent adduct. To remedy this damaging mechanistic feature, structure-guided engineering was performed aimed at preventing the formation of this rate-limiting enzyme intermediate. Two residues were flagged as crucial in determining the fate of the reaction: D151 and Q425. By introducing two more mutations (D151E and Q425S), a variant was created that combined all desired features (stability, selectivity and activity), acting as an efficient 4-*n*-

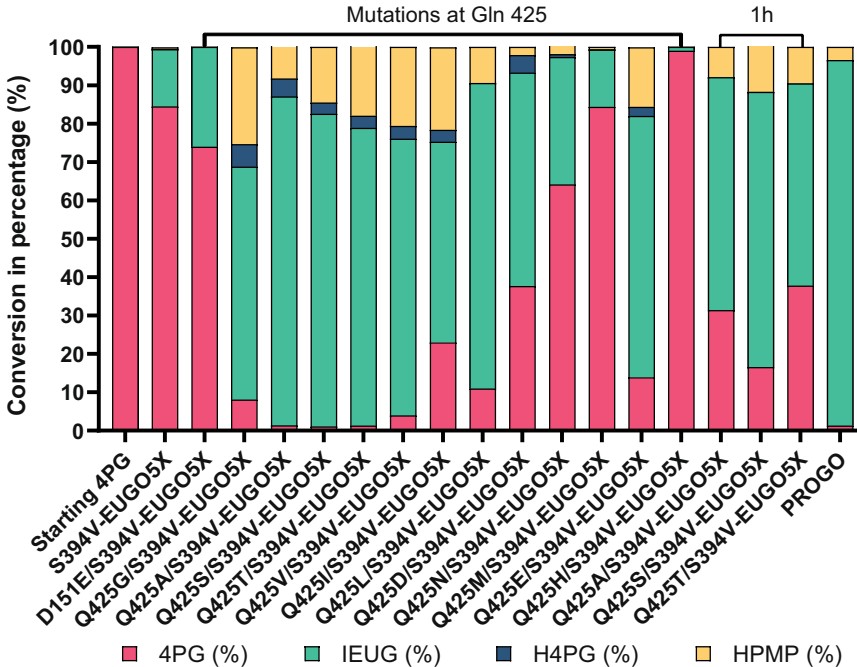

**Fig. 6 | 4-n-Propylguaiacol conversion by purified EUGO5X mutants.** 4-n-propylguaiacol (4PG), isoeugenol (IEUG), 4-(1-hydroxypropyl)−2-methoxyphenol (H4PG), 1-(4-hydroxy-3-methoxyphenyl)−1-propanone (HPMP) are shown in red, green, orange and blue, respectively. Conversions of 4-n-propylguaiacol (5 mM) were carried out in the presence of enzyme (10 μM) in 50 mM KPi, 10% v/v DMSO at pH 7.5 for 3 h, except for the three indicated 1 h conversions.

propylguaiacol oxidase (PROGO). These results show that a stepwise approach, using structural and mechanistic insights and computational guidance, screening of relatively small "smart" mutant libraries, a flavoprotein oxidase can be tuned into a robust enzyme acting on a non-natural compound. In fact, only 104 single-site mutants needed to be expressed and tested in order to identify the eight mutations required to confer enhanced stability, chemoselectivity, and activity. The generated biocatalyst effectively converted 4-n-propylguaiacol into isoeugenol at gram-scale, using isolated enzyme or whole cells. This illustrates the potential of the engineered oxidase in biocatalytic valorization of lignin-derived compounds.

## Methods
### Materials, strains, and chemicals
Oligonucleotide primers for mutagenesis were purchased from Sigma-Aldrich. PfuUltra II HotStart PCR (Polymerase) master mix from Agilent Technologies was used for generating the QuickChange mutations. QIAprep Spin Miniprep Kit from QIAGEN was applied to extract plasmids, and DNA sequencing was provided by Eurofins Genomics. 96-well plates for cell growth, expression, and purification were purchased from Pall Corporation. SYPRO™ Orange protein stain (5000× concentrated in DMSO) was obtained from ThermoFisher Scientific. All other chemicals and reagents in this study were acquired from Sigma-Aldrich or TCI Europe.

The expression plasmid *pBAD-EUGO-His6x* was used, resulting in the expression of EUGO (UNIPROT:Q0SBK1) fused with a C-terminal His-tag through induction by L-arabinose. Unless stated otherwise, all the single point mutants and combined mutants were prepared using *pBAD-EUGO-His6x* as the basis. *Escherichia coli* NEB10β (New England Biolabs) was used as the host strain to express recombinant EUGO.

Purification of isoeugenol was performed by column chromatography using Merck 60 Å 230−400 mesh silica gel. NMR data were collected on a Varian Mercury Plus ($^1$H at 400 MHz and $^{13}$C at 101 MHz) equipped with a 5 mm PFG AutoSW probe. Chemical shifts are reported in parts per million (ppm) relative to the residual solvent peak (CDCl$_3$, $^1$H: 7.26 ppm). Coupling constants are reported in Hertz (Hz).

### Improving enzyme stability with FRESCO
The structure of EUGO available as PBD entry 5FXD[29] was prepared in Yasara[38] according to the FRESCO protocol[33]. The ligands were removed and the protonation state of the structure was optimized. The structure was kept in its dimeric state. To avoid disturbing the binding of the FAD cofactor by mutating proximal residues, a file containing a list of allowed mutations was prepared. This list included all residues having atoms further than 5 Å from the FAD. The calculations were performed for all 9196 allowed mutations using FoldX[39] and Rosetta[40].

In the following step, only mutations predicted by FoldX and Rosetta to improve stability by at least 5 kJ mol$^{-1}$ in calculated ΔΔG were selected. After preparing the selected pool of 496 mutants by adding water molecules, MD simulations were performed. The structures from MD simulations were visually inspected, and mutants were accepted or discarded according to two principal criteria: (1) The quality of the physicochemical interactions between the new residues and their environment. For example, new mutations showing hydrophobic exposure were discarded, and mutations to polar residues forming new hydrogen bonds were accepted, as well as mutations to bulkier hydrophobic residues that were filling available space into a hydrophobic core. (2) The structural variability upon mutation. A spatial superposition of the structures found after the 5 MD simulations per variant, allowed us to depict, on the one hand, if the predicted interactions were maintained in time, and on the other hand, the degree of structural variability upon mutation. All the mutations showing a high structural variability were discarded. Based on these criteria, and giving a score of 1 to every detrimental effect observed per mutation, only 72 mutations scored 0 and were selected for experimental testing (Supplementary Table 1). Overall, the fact that only one variant every seven was accepted, evidences that the used methodologies for predicting mutations based on the improvement of ΔΔG (Rosetta ΔΔG and FoldX) fail often to predict suitable environments for residues at certain targeted positions, which leaves room for improvement. As observed in previous FRESCO exercises, these appeared distributed all over the protein structure.

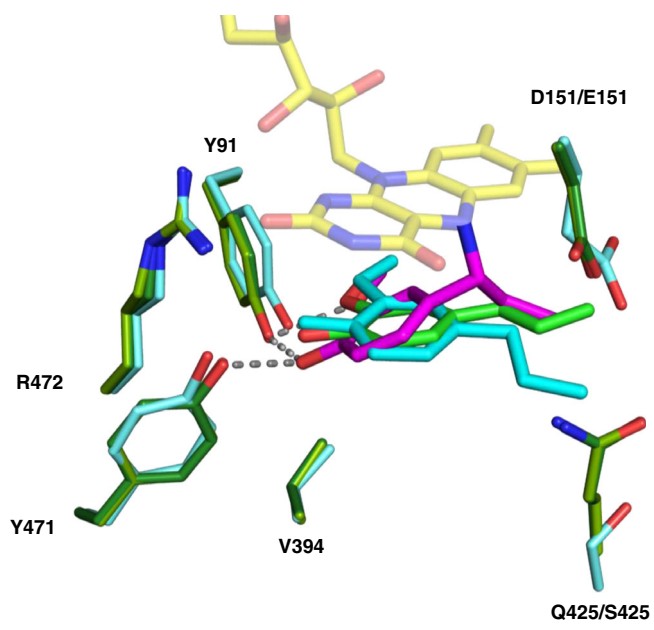

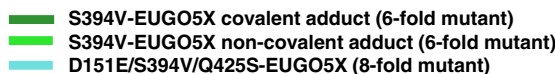

**S394V-EUGO5X covalent adduct (6-fold mutant)**
**S394V-EUGO5X non-covalent adduct (6-fold mutant)**
**D151E/S394V/Q425S-EUGO5X (8-fold mutant)**

**Fig. 7 | Structural comparison between S394V-EUGO5X (covalent and non-covalent) and PROGO.** The interactions with Y91 and Y471 are shown with dashed lines. They are involved in deprotonating 4-*n*-propylguaiacol (Fig. 1). The covalent and non-covalent substrates found in the structure of the S394V-EUGO5X mutant are shown in magenta and green, respectively. The substrate bound to the D151E/S394V/Q425S-EUGO5X mutant (PROGO) is shown in cyan. The side chains of the S394V-EUGO5X mutant are represented in green, with a darker green used for the subunit bearing the covalent adduct. The side chains of D151E/S394V/Q425S-EUGO5X are represented in light blue.

## Mutagenesis, expression, and purification of mutants in 96-well microplates

Following well-established protocols[41], mutant constructs obtained via QuickChange mutagenesis were chemically transformed into *Escherichia coli* NEB 10β competent cells and further confirmed by DNA sequencing. Precultures of mutants were grown in a 96-deep well plate containing 1 mL LB with ampicillin (50 mg/L) per well at 37 °C. In duplicate, another two 96-deep well plates with 1 mL Terrific Broth per well were inoculated with overnight precultures (4% v/v) at 37 °C. After approximately 4 h, 0.02% arabinose was added, and the 96-deep well plates were incubated for 24 h at 30 °C. Cells were harvested in the 96-well plates at 2250 g for 20 min at 4 °C. The single pellet for each mutant was resuspended in 200 μL of lysis buffer (lysozyme 1 mg mL⁻¹, NaCl 150 mM in 50 mM KPi pH 8.0) to incubate with moderate shaking for 30 min at 25 °C, and then, it was frozen in liquid nitrogen or −70 °C for 30 min. The cell-free extract was centrifuged at 2250×*g* for 45 min at 4 °C, and the soluble fraction was filtered by a 96-well microplate (Whatman UNIFILTER 96-well Microplate, GE-Healthcare) and then moved into an AcroPrep Advance 1 mL 96-well plate (Pall) with 100 μL of pre-equilibrated Ni-Sepharose resin (GE-Healthcare) for 30 min incubation. The flow-through was removed, and the column was washed three times with 200 μL of 50 mM KPi at pH 8.0 with 150 mM NaCl and one time with the same buffer containing 5 mM imidazole. The protein was eluted with 100 μL of 50 mm KPi containing 150 mM NaCl and 500 mM imidazole. The eluate was desalted in 50 mM KPi buffer at pH 7.5 using PD MultiTrap G-25 plates (GE-Healthcare).

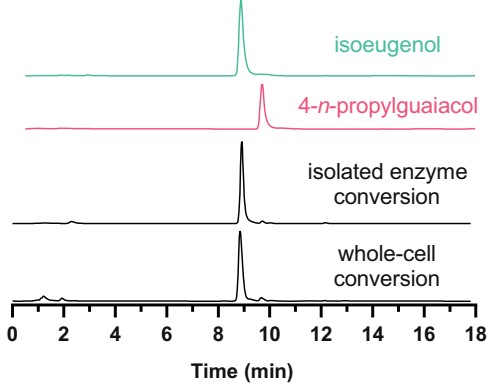

**Fig. 8 | HPLC analysis of the conversion of 4-*n*-propylguaiacol by PROGO.** The first two chromatograms show the elution of the reference compounds, isoeugenol (green) and 4-*n*-propylguaiacol (red). The chromatograms of the conversions (black) show depletion of 4-*n*-propylguaiacol (R$_t$ = 9.7 min) to form isoeugenol (R$_t$ = 8.9 min). The conversions were performed in 48 h at 25 °C using 10% (v/v) DMSO, 50 mM KPi (pH 7.5) as buffer. For the conversion by isolated enzyme, 0.5 gram (3.0 mmol) 4-*n*-propylguaiacol and 18.5 mg (0.30 μmol) purified PROGO was used in a 60 mL reaction volume. For the conversion by whole cells, 1.27 gram 4-*n*-propylguaiacol (7.6 mmol) and *E. coli* cells from a 150 mL culture (final OD$_{600}$ = 29) was used in a 125 mL volume.

## Measuring thermostability of EUGO variants

EUGO is a FAD-containing enzyme which suggests that the ThermoFAD[41] method could be used to determine apparent melting temperatures ($T_m$). However, only a poor signal was obtained during these experiments when tested with wild-type EUGO. This is probably due to the covalent flavin-protein linkage, which causes fluorescence quenching of the flavin, even when the protein is unfolded. The Thermofluor assay was used instead and was performed by mixing 20 μM purified protein with 20 units SYPRO Orange to measure the apparent melting temperature[42]. The relative fluorescence change was recorded from 20 to 99 °C with 0.5 °C increments every 10 s in the CFX96 Touch Real-Time PCR Detection System (Bio-Rad Laboratories).

## Enzyme kinetics

Enzyme activity of EUGO and its mutants on vanillyl alcohol was determined spectrophotometrically (V-660; Jasco) by monitoring the increase of absorbance at 340 nm due to the formation of vanillin ($\varepsilon$ = 14.0 mM⁻¹ cm⁻¹ at pH 7.5). The 500 μL reactions contained 50 mM KPi buffer pH 7.5, 0.05 μM enzyme, and various concentrations of vanillyl alcohol. Likewise, activity on 4-*n*-propylguaiacol was measured by following the absorbance changes at 300 nm, reflecting the formation of isoeugenol. The 500 μL reactions contained 50 mM KPi buffer pH 7.5, 1.0 μM enzyme, 10 μM to 1 mM 4-*n*-propylguaiacol, and 10% v/v DMSO. When analyzing PROGO, the activity on 4-*n*-propylguaiacol was measured with a 100 nM enzyme. After adding the enzyme and quickly mixing, the reaction was followed for 30 s. The steady-state parameters were determined by fitting the data using the Michaelis–Menten equation (GraphPad Prism 6).

To examine the apparent redox state of oxidase variants during and after conversion, flavin absorbance spectra (300–700 nm) were recorded. For this, an aerated solution of 200 μM 4-*n*-propylguaiacol was mixed with 10 μM S394V-EUGO5X or PROGO, respectively. Spectra were collected at regular time intervals.

## Smart library for chemoselectivity of EUGO on 4-*n*-propylguaiacol

**In silico active site mutagenesis.** The X-ray structure of the WT EUGO bound to eugenol (PDB: 5FXD) was used as a template. The structure

was first cleaned with Yasara, the hydrogen atoms were added, and the hydrogen bond networks were optimized[43]. The 4-n-propylguaiacol substrate was included in the active site of EUGO using Autodock VINA[35]. The eugenol present in the X-ray structure was used to define the center of the simulation box, comprising 3 Å from the ligand, and then it was manually removed. The best scored binding mode of 4-n-propylguaiacol at the active site was used as starting point for active site engineering. The 4-n-propylguaiacol substrate docked into the active site was used to define the area for Rosetta engineering, comprising 5 Å from the ligand. Fourteen non-catalytic active site residues were selected for mutagenesis (Fig. 3a) and were allowed to mutate to any residue found at a specific position with more than 2% of the population in a multiple sequence alignment (1001 sequences grouped in 14 clusters) performed with ClustalO[44]. A Monte Carlo algorithm explored individual changes resulting in a library of single variants. Repacking was allowed for all the residues in contact with any modified position through the flag-exclude nonclashing positions. A Boltzmann constant of 0.6 and ligand weight of 1.0 were used. Every individual experiment consisted of 50 Coupled Moves runs of 1000 trials. All the resulting variants were assessed by protein–ligand docking and the distances between the hydroxyl group of 4-n-propylguaiacol and H91 and H471 (<3.5 Å) and binding energies (ΔG) were considered to be similar or better than the wild-type structure. The resulting 16 variants were selected for experimental validation (Fig. 3b). For analysis and presentation of the data, Weblogo[45], Matplotlib 3.6.2[46], and Seaborn[47] were used.

**Experimental validation.** Recombinants of single mutations and combined mutations were routinely prepared by QuickChange mutagenesis as described above. To screen this library quickly and easily, cells were grown in 10 mL Terrific Broth, induced to express the mutant enzymes, and harvested after 24 h. Supernatants of cell-free extract were acquired by sonicating the cells resuspended in 1 mL lysis buffer containing 50 mM KPi, 5% v/v glycerol, and 500 mM NaCl, pH 8.0 and centrifuged for 30 min, 18514×g at 4 °C. The 4-n-propylguaiacol conversions were performed with 10× diluted cell-free extracts or 10 μM purified enzyme, and analyzed by HPLC using a C8 column with a 25–95% gradient acetonitrile for 12 min. All the single mutants in the library were first screened using cell-free extracts. Best hits from this round were further tested by using purified mutant enzymes (10 μM). To determine the dehydrogenating activity, the products' yields were quantified. Calibration curves of 4-n-propylguaiacol, 4-(1-hydroxypropyl)−2-methoxyphenol, 1-(4-hydroxy-3-methoxyphenyl)−1-propanone and isoeugenol were established in the range from 0.01 mM to 10 mM.

### Expression and purification of S394V-EUGO5X and D151E/S394V/Q425S-EUGO5X mutants
The expression of the mutants was performed by growing single colonies (*pBAD-His-Sumo-S394V-EUGO5X* and *pBAD-His-SUMO-D151E/S394V/Q425S-EUGO5X*) in LB medium supplemented with ampicillin (50 μg mL⁻¹) at 37 °C overnight. The precultures were then transferred to 1 L Terrific Broth cultures (1:100) and grown at 37 °C, 11×g until the O.D reached 0.6–0.7. The cultures were then induced with 0.02% w/v of L-arabinose, and cells were grown for 20 h at 30 °C. Cells were harvested by centrifugation (6000×g, 15 min, 10 °C) and the pellet was resuspended in lysis buffer (50 mM Tris-HCl pH 8.0, 150 mM NaCl, 5 mM imidazole, 1 mg mL⁻¹ lysozyme, 10 μM FAD), including additional protease inhibitors−phenylmethylsulfonyl fluoride (1 mM), leupeptin (10 μM), pepstatin (10 μM)−and 1 mg DNase I per 50 mL. Cell lysis was conducted by sonication using the following condition: pulse 5 s on, 25 s off, with a total sonication time of 2 min and 30% amplitude. Lysed cells were centrifuged (56,000×g, 1 h, 4 °C), and the supernatant was collected and filtered (0.45 μm) prior to being loaded onto the HisTrap HP column (5 mL of resin, Cytiva), pre-equilibrated with Buffer A

(50 mM Tris-HCl pH 8.0, 150 mM NaCl, 5 mM imidazole). The His-tagged protein was eluted with elution buffer (50 mM Tris-HCl pH 8.0, 150 mM NaCl, 300 mM imidazole) and concentrated to a final volume of 1 mL. Subsequently, the sample was incubated with 6xHis-tagged SUMO protease (1.1 mg mL⁻¹) to a volume ratio of 1:100 and dialyzed overnight using a 10k dialysis cassette (ThermoFisher) to remove imidazole. After buffer exchange, the protein was then loaded onto a HisTrap column (5 mL, Cytiva) to perform a reverse-nickel purification. The column was pre-equilibrated with Buffer A, with the protein eluting immediately. The tag-less protein was concentrated to a final volume of 500 μL and incubated with 1 mM FAD overnight at 4 °C. The day after, the sample was loaded onto a gel filtration column (Superdex 200 10/300, Cytiva) pre-equilibrated with 10 mM Tris-HCl, pH 7.5 at 4 °C. The protein was eluted with very high purity and homogeneity, and an elution volume of 11.5−12 mL.

### Crystallization of S394V-EUGO5X and D151E/S394V/Q425S-EUGO5X mutants
Purified S394V-EUGO5X was concentrated to 27 mg mL⁻¹ in 10 mM Tris-HCl buffer pH 7.5 at 4 °C. Crystallization was performed using the vapor-diffusion sitting-drop technique at 20 °C by mixing equal volumes of protein and precipitant solution consisting of 0.2 M MgCl₂, 0.1 M Tris pH 8.5, 3.4 M 1,6-hexanediol. After two weeks, thin and squared yellow crystals were obtained. They were soaked for 1 h in a cryoprotected solution consisting of 0.2 M MgCl₂, 0.1 M Tris pH 8.5, 3.6 M 1,6-hexanediol and 5 mM 4-n-propylguaiacol. Crystals of D151E/S394V/Q425S-EUGO5X mutant were grown at 20 °C by the sitting-drop vapor-diffusion method. Protein (15 mg mL⁻¹) was mixed with an equal volume of a precipitant solution consisting of 0.06 M of MgCl₂·6H₂O/CaCl₂·2H₂O, 0.1 M sodium HEPES/MOPS (acid), 20% ethylene glycol, 20% (w/v) PEG 8000. After 1 week, one hexagonal yellow crystal was obtained and soaked for 1 h in the reservoir solution with 5 mM 4-n-propylguaiacol. X-ray diffraction data used for structure determination and refinement were collected at the PXI beamline of the Swiss Light Source in Villigen (SLS, Switzerland) and at Massif3 beamline of the European Synchrotron Radiation Facility (ESRF, Grenoble). The crystal structures were solved by molecular replacement using the coordinates of EUGO from *Rhodococcus jostii* RHA1 (PDB entry 5FXD) as a search model excluding the ligand and water molecules. Crystallographic computing, manual building, the addition of water, and crystallographic refinement were performed with COOT[48] and Phenix[49]. Figures were created with PyMOL (DeLano Scientific; www.pymol.org) and Chimera[50]. Crystallographic statistics are listed in Supplementary Table 3.

### Reporting summary
Further information on research design is available in the Nature Portfolio Reporting Summary linked to this article.

## Data availability
The structural coordinates generated in this study have been deposited in the Protein Data Bank under accession codes: 7YWU and 7YWV. The structural coordinates of wild-type EUGO (5FXD) are available from the Protein Data Bank. All data and materials supporting the findings in the manuscript are available in the Supplementary Information and the Source Data file and from the corresponding author upon request. Source data are provided with this paper.

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

## Acknowledgements

This project has received funding from the Bio-based Industries Joint Undertaking (JU) under the European Union's Horizon 2020 research and innovation program under grant agreement No. 837890. The JU receives support from the European Union's Horizon 2020 research and innovation program and the Bio-based Industries Consortium. B-Ligzymes (GA 824017) from the European Union's Horizon 2020 Research and Innovation Program is also acknowledged for funding the secondment of L.A.C. at the University of Groningen.

## Author contributions

Y.G. performed most biochemical experiments and wrote the first draft of the manuscripts. L.A. crystallized, refined EUGO variants, analyzed the structure, and wrote the corresponding part of the manuscript. M.T. contributed to experimental work, L.A.C. and E.M. performed computational analyses, and S.S. performed the FRESCO analysis. I.M. purified the isoeugenol from preparative scale conversions and characterized it with NMR. M.W.F. and A.M. initiated the project and led the project team, designed the experiments, and guided all aspects of this study.

## Competing interests

The authors declare no competing interests.
