## [Peer Review File · Nature Communications]

Structure- and computational-aided engineering of an oxidase to produce isoeugenol from a lignin-derived compoundREVIEWER COMMENTS

Reviewer #1 (Remarks to the Author):

This manuscript focused on the development of a novel 4-n-propylguaiacol oxidase according to the engineering the eugenol oxidase from *Rhodococcus jostii* RHA124. After engineering a thermostable variant through computational design, the required catalytic activity was introduced by a multi-step computational- and structure-guided enzyme engineering strategy. This resulted in an engineered 4-n-propylguaiacol oxidase (PROGO) which is stable and efficiently oxidizes 4-n-propylguaiacol into isoeugenol with good activity, chemical selectivity and thermo-stability. The current work is meaningful and innovative. The manuscript can be accepted after revision.

1. Page 7, line 169-171: For the construction of EUGO's chemoselective smart library, the selection of other 10 mutation sites near the active center needs more explanation.
2. Page 9, line 254 and Page 28, figure 6: From figure 6, Q425A showed higher selectivity of IEUG and similar conversion compared with Q425S. And it may also be a good candidate to combine with D151E/S394V-EUGO5X. Have you tested this mutant combination?
3. A preparative scale experiment for the synthesis of isoeugenol is necessary to explore the practical application potential of D151E/S394V/Q425S-EUGO5X mutant.
4. Page 29, line 652: "Tyr 47" should be modified to "Tyr 471".

Reviewer #2 (Remarks to the Author):

The manuscript by Guo et al. describes the engineering of eugenol oxidase (EUGO) to catalyze the dehydrogenation of 4-n-propylguaiacol into isoeugenol. The authors first substituted a number of residues to increase the thermostability of the enzyme, then introduced three additional substitutions into the thermostable variant to increase the specificity and selectivity of the enzyme for the dehydrogenation of 4-n-propylguaiacol. The substitutions were selected based on a combination of computational and structure-guided approaches. For the most part, the manuscript is well-written, the data are clearly presented, and the data are convincing. Nevertheless, some additional experiments would strengthen the conclusions and broaden the significance of the work.

Specific comments:

1. The manuscript is quite well-written and is relatively free of grammatical and stylistic errors. However, English throughout the manuscript should be improved. The following are cited as examples; the list is not exhaustive:

"This study aimed at engineering a VAO-type oxidase capable of selectively dehydrogenating 4-n-propylguaiacol into isoeugenol." could be "This study aimed to engineer a VAO-type oxidase to catalyze the selective dehydrogenation of 4-n-propyl-guaiacol to isoeugenol" (ll. 101-102).

"in an area of 5 Å from" should be "within 5 Å of" (l. 162)

"to avoid that the expression and/or folding of the protein was compromised" should be "to avoid perturbing the expression and/or folding of the protein" (ll. 166-167)

"Noticeably" should be "Notably" (l. 209)

"it merely requires oxygen as mild oxidant" should be "it requires only oxygen as a co-substrate" (l. 273)

2. In describing isoeugenol as a valuable chemical to target, the authors should state the global market (in dollars) for isoeugenol (p. 3).

3. In screening variants in cell lysates for conversion of 4-n-propylguaiacol, how were different amounts of variants in the lysate taken into account (p. 7)?
4. The covalent adduct is variously referred to as an “unwanted inactivating side reaction” (l. 1.06) and a “(rate-limiting) enzyme intermediate” (l. 226, l. 292). The authors should use more consistent terminology and, at some point in the text, clearly describe the significance of the adduct with respect to the enzyme’s proposed catalytic mechanism. That is, is the adduct a catalytic intermediate or the result of an irreversible, unproductive side reaction?
5. The catalytic properties of D151E/S394V-EUGO5X should be determined and reported in Table 2. This variant was a key intermediate between S394V-EUGO5X and the final engineered enzyme, PROGO, so it would be informative to know this variant’s kcat value for 4-n-propylguaiacol.
6. For KM and kcat values, state the total number of data points used in their determination, not the number of “replicates” (l. 580).
7. The Conclusion should be revised to minimize repetition of the Abstract and the last paragraph of the Introduction.
8. The authors conclude that PROGO is a “robust oxidase will enable biocatalytic valorization of lignin-derived compounds” (l. 298). However, the direct relevance of the experimental conditions (e.g., 5 mM 4-n-propylguaiacol, 50 mM phosphate, pH 7.5, 3 hours) to a biocatalytic application are unclear. The authors should demonstrate the utility of the enzyme for biocatalysis under more relevant conditions, even if only on a small scale.

Reviewer #3 (Remarks to the Author):

4-n-propylguaiacol may be derived from lignin, but biocatalysts are not known capable of dehydrogenation of 4-n-propylguaiacol into isoeugenol. The authors engineer eugenol oxidase to catalyse the desired reaction. First, thermostability of the enzyme is improved which is followed by finetuning of the catalytic activity. For enzyme engineering authors have utilized the structural knowledge, and computational and intuitive design methods to arrive at an engineered enzyme characterized by 97% chemoselectivity.

Although the study is technically sound, it adds little to our general ability of enzyme repurposing / redesign or actual capability of obtaining useful compounds from lignin in industrial setting. The methods reported for enzyme stabilization and activity redesign, though eventually successful, are not novel or directly suitable for general application, involve a lot of guesswork and unpredictability and as such do not constitute a significant progress in the field. Likely, AI approaches will at one point provide such a breakthrough, but this awaits implementation. Despite those reservations, the procedure allows the authors to arrive at D151E/S394V/Q425S-EUGO5X, a mutant of reportedly desired properties. However, the turnover rate of such mutant for 4-n-propylguaiacol is an order of magnitude less than that of the WT for vanillyl alcohol suggesting that the mutant may accommodate yet significant improvement of parameters. Even more pronounced estimate of the room for improvement is provided by catalytic efficiency (kcat/kM; which is by the way neither reported nor discussed in the text) which one can calculate from data provided in Table 2 at $\sim 5.8 \times 10^4 \text{ M}^{-1} \text{ s}^{-1}$ while in principle values exceeding 1.0×10^8 are theoretically possible while values exceeding 1.0×10^6 are not uncommon. In the lack of demonstration of the direct usefulness of the obtained mutant in efficient processing of lignin derived 4-n-propylguaiacol in the pilot settings it is impossible to conclude after the authors that the mutant will “enable biocatalytic valorisation of lignin-derived compounds” which further undermines the overall value of the outcome of the study.

Of the detailed comments:

1. The abstract should be rewritten to remove repetitive statements.
2. In the Introduction section the authors introduce 4-n-propylguaiacol as a possible (conditions depending) dominant product of the RCF process. 4-n-propylguaiacol may be converted to isoeugenol by the process described in the manuscript, while the latter may find a number of uses to

obtain products described in the Introduction. It would be advantageous if the authors estimated and stated the demand for such products to let the readers appreciate the potential value of the elaborated process, i.e. the volume of use of isoeugenol as flavour and fragrance, the estimated volume of consumption of vaniline, polymers and epoxy resins, etc. And following, in lines 51 and 52 the authors call isoeugenol the “valued monophenol”. An estimate of annual consumption of this chemical would be useful.

3. Please justify the reason why ligand bound enzyme is used for stability calculations (126-127). Ligands tend to stabilize the enzymes. Why was the apo form not tested in parallel?

4. Please, define more strictly the criteria used for “visual inspection” (130 and materials and methods). This process selects 1 in seven tested mutations (72 out of 496). Such a great limitation of the test set should be better justified.

5. Out of 70 tested mutants 45 are neutral, 16 have the effect opposite to the expected effect and only 7 show significant improvement in thermal stability. The success rate seems relatively low. How does it compare to semirandom mutagenesis in which one introduces non-nonsense mutations at random sites? (by non-nonsense mutations I understand mutations which are not visibly incompatible with the structure (i.e. substitution of Ala with Trp at the tightly packed core of the protein). Please, benchmark the FRESCO prediction method.

6. Please, describe the rationale behind the expectation that combining beneficial mutations will produce a beneficial effect (147-152). It obviously does in this case, but one can easily imagine that two beneficial mutations cancel each-other effects. How were the order of mutation addition chosen? Another order could result in different intermediate results. The reviewer appreciates the positive final result of the FRESCO approach; however, the method contains a significant random component (comment 5 and this comment). May this be converted to a more algorithmic approach?

7. Line 156 – please indicate that the number in parentheses refers to kcat.

8. Enzyme repurposing – the authors present a semi algorithmic procedure 159-167 for selection of potentially preferable mutations, but then supplement the list of mutants with additional mutants (169-171) designed on poorly justified basis, as if implying that the earlier described algorithm is not sufficiently trustworthy. Such reservation seems confirmed by the fact that out of 26 tested mutations most were compromised in terms of conversion level while the effect on chemoselectivity is also widely variable. The approach, though eventually successful is inefficient in terms of predictive value of the computational/design steps.

9. In line 206 the authors explain the stabilizing effect of the mutation with the statement that “substitutes a hydrophobic residue with a charged one on the protein surface”. It is far from obvious to me why such a mutation would stabilise the protein. It may affect solubility, but why thermal stability? Could the authors cite a reference to a broader study investigating the effect of surface mutations on protein stability? Or otherwise provide a more reliable explanation.

10. Figure 5 panel A – the view at the cavity should be enlarged. It is difficult to see the discussed features of the active site at this panel.

11. 208-214 – It is difficult to discern in Figure 5 the water channel described in the text. Please indicate the channel on the figure with different colouring.

12. 231-237 is discussing a covalent adduct formation between flavin and the substrate. The text redirects the reader to figure 5, but figure 5 does not show evidence of the covalent bond formation. Interpreting a difference between covalent and noncovalent binding at 2.8 angstrom resolution is a difficult (if not impossible) undertaking. The authors should at least show a figure explaining the claim that they can discern the covalent and noncovalent binding in the electron density describing different molecules in ASU. Supporting evidence of covalent bond formation (i.e., from MS) would be useful. Apart from that, how do the authors explain that some molecules in ASU contain covalently bound product and some not, instead of all of the protein molecules containing averaged superposition of densities describing covalent and noncovalent states? While would a crystal select covalently and noncovalently bound species at different sites in ASU instead of averaging out the distributions.

13. Were the subunits which are described in 235 and onwards compared after fixing the covalent bond in some subunits and not in the others? If so, the described tilting may be an effect of the imposed restraints and not of true differences between observed electron densities. It would be beneficial to compare not the models, but rather the densities in the omit maps in the regions of interest between all 8 subunits and report some statistical evaluation of the significance of observed differences.

14. 252-253 – Please specify which mutants the authors have in mind. What is considered the “additional mutation” here and what is the “parent mutant”

15. The authors mention (257-258) that the mutants “kcat has improved by two orders of magnitude when compared with wild type EUGO” however fail to notice that this is still an order of magnitude below the kcat of the wild type for vanillyl alcohol, that is, there is still potential for turnover improvement. This should be included in the discussion.

16. 262 and figure 7 – please show the water channels, Which direction is the water accessing, from where?

17. 262-267 – the description of the effect of structural changes on catalysis is a bit too “naive” to me. Explaining covalent bond formation by molecular crowding seems an oversimplification. The authors should provide simulations supporting such conclusion.

18. Rather than repeating the results, the discussion section should discuss the obtained results against the landscape of current knowledge. Are there other/better methods of mutant prediction available? How does FRESCO benchmarks against those? Were the difficulties encountered in this study similar to other experience in enzyme repurposing. Are the qualities of the obtained enzyme sufficient for industrial utilization? Does it truly have a chance to “enable biocatalytic valorisation of lignin-derived compounds” (289). Etc. – there are a lot of topics one could focus the discussion at, instead of simply summarizing the results.

RESPONSE TO COMMENTS

Reviewer #1 (Remarks to the Author):

This manuscript focused on the development of a novel 4-n-propylguaiacol oxidase according to the engineering the eugenol oxidase from *Rhodococcus jostii* RHA124. After engineering a thermostable variant through computational design, the required catalytic activity was introduced by a multi-step computational- and structure-guided enzyme engineering strategy. This resulted in an engineered 4-n-propylguaiacol oxidase (PROGO) which is stable and efficiently oxidizes 4-n-propylguaiacol into isoeugenol with good activity, chemical selectivity and thermo-stability. The current work is meaningful and innovative. The manuscript can be accepted after revision.

1. Page 7, line 169-171: For the construction of EUGO's chemoselective smart library, the selection of other 10 mutation sites near the active center needs more explanation.

> Thanks for this remark. In fact, we just wanted to be complete in reporting on the work done. In the engineering step to improve the chemoselectivity, we had 16 mutations suggested by computational analysis, while we were also curious to see how some other active site mutants suggested by visual inspection of the active site would perform. In fact, none of these 10 mutants did improve the chemoselectivity; most of them resulted in poorly active enzyme. None of the 10 mutations was included in the next round of engineering. Therefore, the revised version does not report on them anymore as they were not part of the computational approach that we report in this study.

2. Page 9, line 254 and Page 28, figure 6: From figure 6, Q425A showed higher selectivity of IEUG and similar conversion compared with Q425S. And it may also be a good candidate to combine with D151E/S394V-EUGO5X. Have you tested this mutant combination?

> The conversion data of the Q425X/S394V mutants revealed that the Q425S mutant was the most efficient in conversion. For clarity, Figure 6 now also includes 1 h conversions for some relevant mutants to further confirm the superior performance of the Q425S/S394V mutant. Therefore, we focused on the multiple mutant harboring the Q425S mutation, and did not test the Q425A mutation any further.

3. A preparative scale experiment for the synthesis of isoeugenol is necessary to explore the practical application potential of D151E/S394V/Q425S-EUGO5X mutant.

> We have performed two experiments: a conversion of 0.5 gram of 4-n-propylguaiacol using our purified PROGO and a conversion of 1.27 gram of 4-n-propylguaiacol using whole cells. In both cases complete conversion is achieved. After conversion, product has been isolated and verified by NMR. The data have been added to the manuscript. With this, we feel that we have demonstrated the capability of the engineered enzyme to be used as biocatalyst for the preparation of isoeugenol, starting from a lignin-derived compound.

4. Page 29, line 652: "Tyr 47" should be modified to "Tyr 471".

> This has been corrected.

Reviewer #2 (Remarks to the Author):

The manuscript by Guo et al. describes the engineering of eugenol oxidase (EUGO) to catalyze the dehydrogenation of 4-n-propylguaiacol into isoeugenol. The authors first substituted a number of residues to increase the thermostability of the enzyme, then introduced three additional substitutions

into the thermostable variant to increase the specificity and selectivity of the enzyme for the dehydrogenation of 4-n-propylguaiacol. The substitutions were selected based on a combination of computational and structure-guided approaches. For the most part, the manuscript is well-written, the data are clearly presented, and the data are convincing. Nevertheless, some additional experiments would strengthen the conclusions and broaden the significance of the work.

Specific comments:

1. The manuscript is quite well-written and is relatively free of grammatical and stylistic errors. However, English throughout the manuscript should be improved. The following are cited as examples; the list is not exhaustive:

“This study aimed at engineering a VAO-type oxidase capable of selectively dehydrogenating 4-n-propylguaiacol into isoeugenol.” could be “This study aimed to engineer a VAO-type oxidase to catalyze the selective dehydrogenation of 4-n-propyl-guaiacol to isoeugenol” (ll. 101-102).

> This has been corrected.

“in an area of 5 Å from” should be “within 5 Å of” (l. 162)

> This has been corrected.

“to avoid that the expression and/or folding of the protein was compromised” should be “to avoid perturbing the expression and/or folding of the protein” (ll. 166-167)

> This has been corrected.

“Noticeably” should be “Notably” (l. 209)

> This has been corrected.

“it merely requires oxygen as mild oxidant” should be “it requires only oxygen as a co-substrate” (l. 273)

> This has been corrected.

2. In describing isoeugenol as a valuable chemical to target, the authors should state the global market (in dollars) for isoeugenol (p. 3).

> Isoeugenol is known from various applications, as indicated in the revised manuscript. A documented example of its widespread usage is in deodorants (new reference 7). We have now added also a recent paper in which it is demonstrated that isoeugenol can be efficiently converting into vanillin using an engineered isoeugenol dioxygenase. For vanillin, the estimate market value was 480 million USD in 2019 (new reference 8). With this engineered biocatalyst, a biotechnological process for producing vanillin from lignin is in reach.

3. In screening variants in cell lysates for conversion of 4-n-propylguaiacol, how were different amounts of variants in the lysate taken into account (p. 7)?

> We tried to normalize the expression conditions as much as possible, and observed very similar expression levels. Most importantly, we selected mutants that displayed improved chemoselectivity, which does not depend on the amount of enzyme. In the later stage of enzyme engineering efforts, we used defined amounts of purified enzyme to establish which mutation improve the catalytic performance. This assures that we identify the mutations that improve activity.

4. The covalent adduct is variously referred to as an “unwanted inactivating side reaction” (l. 1.06) and a “(rate-limiting) enzyme intermediate” (l. 226, l. 292). The authors should use more consistent terminology and, at some point in the text, clearly describe the significance of the adduct with respect to the enzyme’s proposed catalytic mechanism. That is, is the adduct a catalytic intermediate or the result of an irreversible, unproductive side reaction?

> As requested, to be more consistent, we have rephrased some of the text that refers to the observed covalent adduct. Furthermore, we have performed additional experiments to verify whether or not formation of the adduct is reversible. This has revealed that prolonged incubation of the S394V-EUGO5X mutant leads to a fully reoxidized flavin cofactor. This information (adduct formation is not irreversible) has been added (p. 9 and Figure S1). The type of covalent adduct (a flavin N5-adduct) and kinetic behavior is highly similar to the adduct previously observed in a structurally and mechanistically related oxidase, vanillyl-alcohol oxidase (a p-cresol-FAD adduct in that case). Even the crystallographic data point to a similar adduct. We have referred to previous work on the adduct observed in VAO (ref #33).

5. The catalytic properties of D151E/S394V-EUGO5X should be determined and reported in Table 2. This variant was a key intermediate between S394V-EUGO5X and the final engineered enzyme, PROGO, so it would be informative to know this variant’s kcat value for 4-n-propylguaiaicol.

> In the third and last phase of the enzyme engineering approach, we focused on mutations that would result in a variant with improved activity. From the data in Figure 6 it can be noted that the various 7-fold ‘intermediate’ mutants (13 in total, including D151E/S394V-EUGO5X) have inferior performances compared with the 8-fold mutant. Therefore, we did not analyze in detail intermediate mutations (such as D151E/S394V-EUGO5X) because they showed poor performance. Only by combining Q425S and D151E a well-performing oxidase was found. On this basis, we performed a full kinetic and structural analysis of the 6-fold and 8-fold (PROGO) mutants for which we report on the stabilities, kinetics (Table 2) and crystal structures.

6. For KM and kcat values, state the total number of data points used in their determination, not the number of “replicates” (l. 580).

> This has been corrected. A total of 7 substrate concentrations were used, and for each concentration two measurements were performed.

7. The Conclusion should be revised to minimize repetition of the Abstract and the last paragraph of the Introduction.

> Parts of the abstract, introduction and conclusion sections have been rewritten.

8. The authors conclude that PROGO is a “robust oxidase will enable biocatalytic valorization of lignin-derived compounds” (l. 298). However, the direct relevance of the experimental conditions (e.g., 5 mM 4-n-propylguaiaicol, 50 mM phosphate, pH 7.5, 3 hours) to a biocatalytic application are unclear. The authors should demonstrate the utility of the enzyme for biocatalysis under more relevant conditions, even if only on a small scale.

> See answer to point 3 of Reviewer 1.

Reviewer #3 (Remarks to the Author):

4-n-propylguaiacol may be derived from lignin, but biocatalysts are not known capable of dehydrogenation of 4-n-propylguaiacol into isoeugenol. The authors engineer eugenol oxidase to catalyse the desired reaction. First, thermostability of the enzyme is improved which is followed by finetuning of the catalytic activity. For enzyme engineering authors have utilized the structural knowledge, and computational and intuitive design methods to arrive at an engineered enzyme characterized by 97% chemoselectivity.

Although the study is technically sound, it adds little to our general ability of enzyme repurposing / redesign or actual capability of obtaining useful compounds from lignin in industrial setting. The methods reported for enzyme stabilization and activity redesign, though eventually successful, are not novel or directly suitable for general application, involve a lot of guesswork and unpredictability and as such do not constitute a significant progress in the field. Likely, AI approaches will at one point provide such a breakthrough, but this awaits implementation. Despite those reservations, the procedure allows the authors to arrive at D151E/S394V/Q425S-EUGO5X, a mutant of reportedly desired properties. However, the turnover rate of such mutant for 4-n-propylguaiacol is an order of magnitude less than that of the WT for vanillyl alcohol suggesting that the mutant may accommodate yet significant improvement of parameters. Even more pronounced estimate of the room for improvement is provided by catalytic efficiency (k_{cat}/K_M ; which is by the way neither reported nor discussed in the text) which one can calculate from data provided in Table 2 at $\sim 5.8e4 \text{ M}^{-1}\text{s}^{-1}$ while in principle values exceeding $1.0e8$ are theoretically possible while values exceeding $1.0e6$ are not uncommon. In the lack of demonstration of the direct usefulness of the obtained mutant in efficient processing of lignin derived 4-n-propylguaiacol in the pilot settings it is impossible to conclude after the authors that the mutant will “enable biocatalytic valorisation of lignin-derived compounds” which further undermines the overall value of the outcome of the study.

> The comparison of the catalytic efficiency, as defined by the reviewer being the k_{cat}/K_M , with enzymes that perform catalytic close to diffusion limitation is not fair. The rate of catalysis is dependent on the type of reaction, dependent on the energy barrier, on the number of chemical steps, the binding/release of substrate/product, etc.. With oxidases, the enzyme deals with at least two substrates (target molecule and dioxygen) which already limits the rate of catalysis. This is also seen when looking at the reported k_{cat} values for oxidases; these are typically in the range of 1-100 s^{-1} . Further, and for a similar reason, the activity of an oxidase perfectly optimized for converting 4-n-propylguaiacol may never reach the same rate as with vanillyl alcohol as substrate. These are two different compounds, for which the energy barrier for oxidation (hydride transfer) will be different.

There is also a study by Dan Tawfik in which it is pointed out that catalytically perfect enzymes are actually very unusual. They found that for enzymes involved in secondary metabolism, the median k_{cat}/K_M for the natural substrate is only $7.1 \times 10^4 \text{ M}^{-1} \text{ s}^{-1}$ while the median k_{cat} is only 2.5 s^{-1} . (Figure 2 from: Bar-Even, A., Noor, E., Savir, Y., Liebermeister, W., Davidi, D., Tawfik, D. S., & Milo, R. (2011). The moderately efficient enzyme: evolutionary and physicochemical trends shaping enzyme parameters. *Biochemistry*, 50(21), 4402-4410.)

> A robust biocatalyst does need to be very active. To demonstrate that the engineered biocatalyst can be used, we have performed additional experiments: gram-scale conversions that worked out very well.

> The reviewer also remarks that AI approaches may be more powerful in engineering enzymes in the future. We are aware that AI-based tools are currently in use for, for example, structure predictions. Yet, for informing enzyme engineering for improved catalytic properties, no effective AI approaches exist. This may change in the coming years, but it may also take much longer. It seems that very

accurate structure predictions, that would allow reliable docking of ligands/substrates, is still impossible. And that does not even include dynamic events, such as binding and release events, and reactivity requirements.

It is important to keep in mind that there needs to be enough high quality learning data for AI to give good predictions. In case of protein structure predictions there were 1×10^5 X-ray structures to learn from, all with very many amino acid - amino acid pairs to learn from. Another requirement for the successful AI structure predictions were the huge numbers of homologous sequence alignments in the database that allowed to predict vicinity from co-evolution. Together, this huge amount of data allowed to solve the problem of predicting the structure from primary sequence for any protein that has enough sequence homologs in the database. In case of enzyme catalytic activity versus sequence, the problem is arguably chemically far more diverse and the learning datasets are tiny compared to what was available for structure prediction. The breakthrough that the reviewer expects may therefore never materialize.

Of the detailed comments:

1. The abstract should be rewritten to remove repetitive statements.

> The abstract has been rewritten.

2. In the Introduction section the authors introduce 4-n-propylguaiacol as a possible (conditions depending) dominant product of the RCF process. 4-n-propylguaiacol may be converted to isoeugenol by the process described in the manuscript, while the latter may find a number of uses to obtain products described in the Introduction. It would be advantageous if the authors estimated and stated the demand for such products to let the readers appreciate the potential value of the elaborated process, i.e. the volume of use of isoeugenol as flavour and fragrance, the estimated volume of consumption of vaniline, polymers and epoxy resins, etc. And following, in lines 51 and 52 the authors call isoeugenol the “valued monophenol”. An estimate of annual consumption of this chemical would be useful.

> See answer to point 2 of Reviewer 2.

3. Please justify the reason why ligand bound enzyme is used for stability calculations (126-127). Ligands tend to stabilize the enzymes. Why was the apo form not tested in parallel?

> The available X-ray structure is ligand-bound and then the accuracy of the model is higher by keeping the system as such. Manually removing the ligand could negatively affect the predictions at the stage of the MD simulations.

4. Please, define more strictly the criteria used for “visual inspection” (130 and materials and methods). This process selects 1 in seven tested mutations (72 out of 496). Such a great limitation of the test set should be better justified.

> This has been rewritten (p. 13 and Table 3).

5. Out of 70 tested mutants 45 are neutral, 16 have the effect opposite to the expected effect and only 7 show significant improvement in thermal stability. The success rate seems relatively low. How does it compare to semirandom mutagenesis in which one introduces non-nonsense mutations at random sites? (by non-nonsense mutations I understand mutations which are not visibly incompatible with the

structure (i.e. substitution of Ala with Trp at the tightly packed core of the protein). Please, benchmark the FRESCO prediction method.

The success rate of 10% for FRESCO here is in agreement with results obtained elsewhere. Previously, we found success percentages with FRESCO varying from 10 to 17% (as listed in Table S3 in Meng, Janssen et al., ACS Catalysis, 2020). One reason for the low chance of success per mutation is that the overall approach is to find as many stabilizing mutations as possible, which makes it necessary to also include non-elite mutations that could turn out to be destabilizing. Another main cause of the low success rate appears to be that most positions in a large protein are not important for protein stability since they are not located in the critical regions for stability. In larger proteins, inactivation is typically triggered by local unfolding (review: Wijma, Floor, & Janssen, Current opinion in structural biology, 2013) and thus most mutations have no effect since they are not in a region where the unfolding starts. Also with the current enzyme, most dysfunctional mutations (45/63) were neutral. During stability engineering of larger proteins, it is typically observed that in some particular regions the fraction of stabilizing mutations is much higher, as high as 56% for an interface region for a transaminase (Meng, Janssen, et al, ACS catalysis 2020, table S3), which was apparently critical for its stability. It is worth noting that for random mutations the success rate appears to be < 1%, since typically large libraries have to be screened (typically 10^3 to 10^5 variants) to get similar stabilizations as with FRESCO (for example, Giver, Arnold, et al. PNAS, 1998)

6. Please, describe the rationale behind the expectation that combining beneficial mutations will produce a beneficial effect (147-152). It obviously does in this case, but one can easily imagine that two beneficial mutations cancel each-other effects. How were the order of mutation addition chosen? Another order could result in different intermediate results. The reviewer appreciates the positive final result of the FRESCO approach; however, the method contains a significant random component (comment 5 and this comment). May this be converted to a more algorithmic approach?

> It is indeed possible that beneficial mutations would cancel each other out. We have observed such effects in other enzyme engineering projects. Yet, this typically involved residues located close to each other. In this case, the mutations that resulted in the best improvements (Figure 1) and that were combined in the 5X mutant enzyme were quite far from each other. This can also be seen in Figure 4 that highlights the positions of the mutations.

We can add, from experience, that attempts to computationally predict the combination of a series of mutations do not yet work satisfactorily enough to fully replace the reliable, and mostly fast, method of rationally combining mutations in the lab. While in principle a different set of combinations will give a different outcome, in practice finding a functional combination of mutations is mostly not a problem. One simply starts with the most stabilizing mutation and then stepwise adds mutations to the pool that do not seem to be interfering structurally with the already pooled mutations, as judged from visual inspection.

Many efforts have been made on the FRESCO protocol to automatize the strategy of combining mutations and limit the user-made decisions as much as possible (e.g. measurements of RMSF in longer MD simulations, potential energy calculations between the mutations and their environment with respect to the WT, SASA predictions, etc). Unfortunately, none of these efforts on automatizing the mutant selection step led to a better predictability when compared to experimental data. Regarding sequence-based deep learning methods for prediction of thermostable mutations, although they are arising as a promising alternative to the structure-based predictions, there is no evidence yet on their potential to achieve large temperature increases, as we and others managed to achieve with FRESCO for many proteins.

7. Line 156 – please indicate that the number in parentheses refers to kcat.

> This has been corrected.

8. Enzyme repurposing – the authors present a semi algorithmic procedure 159-167 for selection of potentially preferable mutations, but then supplement the list of mutants with additional mutants (169-171) designed on poorly justified basis, as if implying that the earlier described algorithm is not sufficiently trustworthy. Such reservation seems confirmed by the fact that out of 26 tested mutations most were compromised in terms of conversion level while the effect on chemoselectivity is also widely variable. The approach, though eventually successful is inefficient in terms of predictive value of the computational/design steps.

> See answer to point 1 of Reviewer 1.

9. In line 206 the authors explain the stabilizing effect of the mutation with the statement that “substitutes a hydrophobic residue with a charged one on the protein surface”. It is far from obvious to me why such a mutation would stabilise the protein. It may affect solubility, but why thermal stability? Could the authors cite a reference to a broader study investigating the effect of surface mutations on protein stability? Or otherwise provide a more reliable explanation.

>With larger proteins, like the currently engineered enzyme, thermal inactivation is typically irreversible and this is often due to aggregation after partial unfolding of a region of the enzyme. Thus, a mutation that improves solubility (in the partially unfolded state) can contribute to overall stability by preventing the subsequent aggregation event that leads to the irreversible inactivation of the enzyme. This scenario is common for larger proteins and differs from the more commonly studied situation with small reversibly folding proteins. Such small proteins are often selected for thermodynamic studies because their unfolding is reversible and uncomplicated, and aggregation effects there do not play a role while they do in other proteins. An example of a review where this is discussed is Eijsink, V. G., Bjørk, A., Gåseidnes, S., Sirevåg, R., Synstad, B., van den Burg, B., & Vriend, G. (2004). Rational engineering of enzyme stability. *Journal of biotechnology*, 113(1-3), 105-120. An example of a careful study in which it is revealed that a change in solubility of the unfolded state was responsible for the observed increase in the apparent thermal stability of an enzyme is: Augustyniak, W., Brzezinska, A. A., Pijning, T., Wienk, H., Boelens, R., Dijkstra, B. W., & Reetz, M. T. (2012). Biophysical characterization of mutants of *Bacillus subtilis* lipase evolved for thermostability: Factors contributing to increased activity retention. *Protein Science*, 21(4), 487-497. We added an explanatory sentence (p. 8).

10. Figure 5 panel A – the view at the cavity should be enlarged. It is difficult to see the discussed features of the active site at this panel.

> The panel has been enlarged.

11. 208-214 – It is difficult to discern in Figure 5 the water channel described in the text. Please indicate the channel on the figure with different colouring.

> As highlighted in Fig 5A, there is an entrance cavity coming from the subunit interface, guiding substrate to the substrate binding site. This route of ligand migration was confirmed for the structurally and functionally related oxidase, vanillyl alcohol oxidase (reference #18). Arrows have been added to the figure to indicate the route.

12. 231-237 is discussing a covalent adduct formation between flavin and the substrate. The text redirects the reader to figure 5, but figure 5 does not show evidence of the covalent bond formation.

Interpreting a difference between covalent and noncovalent binding at 2.8 angstrom resolution is a difficult (if not impossible) undertaking. The authors should at least show a figure explaining the claim that they can discern the covalent and noncovalent binding in the electron density describing different molecules in ASU. Supporting evidence of covalent bond formation (i.e., from MS) would be useful. Apart from that, how do the authors explain that some molecules in ASU contain covalently bound product and some not, instead of all of the protein molecules containing averaged superposition of densities describing covalent and noncovalent states? While would a crystal select covalently and noncovalently bound species at different sites in ASU instead of averaging out the distributions.

13. Were the subunits which are described in 235 and onwards compared after fixing the covalent bond in some subunits and not in the others? If so, the described tilting may be an effect of the imposed restraints and not of true differences between observed electron densities. It would be beneficial to compare not the models, but rather the densities in the omit maps in the regions of interest between all 8 subunits and report some statistical evaluation of the significance of observed differences.

> Concerning comments 12 & 13: The asymmetric unit of the crystals used for the structural analysis contains eight crystallographically independent subunits. The covalent bond between the substrate and the N5 atom of the FAD cofactor is clearly visible in two of these protein chains. Despite the medium resolution of the crystal (2.8 Å), the stretch of electron density connecting the ligand to the flavin was very well-defined even before inclusion of the ligands in the refinements (i.e. fully unbiased maps). The presence of crystallographically independent subunits in different conformational, catalytic and/or ligation states is not uncommon in crystallography. In our lab, we have had several cases. An historical example is the allosteric lactate dehydrogenase whose crystals contain both R- and T-state subunits (Iwata et al. T and R states in the crystals of bacterial L-lactate dehydrogenase reveal the mechanism for allosteric control. *Nat Struct Biol.* 1994 1:176. doi:10.1038/nsb0394-176). Because of different crystal contacts, the crystalline protein chains might differ with regard to the rates of catalysis, substrate diffusion, or product release. The revised manuscript now comprises figures (Fig. 5b) that shows in more detail the unbiased ligand densities and discusses this feature.

14. 252-253 – Please specify which mutants the authors have in mind. What is considered the “additional mutation” here and what is the “parent mutant”

> The additional mutation refers to the D151E mutation. This has now been indicated in the text.

15. The authors mention (257-258) that the mutants “kcat has improved by two orders of magnitude when compared with wild type EUGO” however fail to notice that this is still an order of magnitude below the kcat of the wild type for vanillyl alcohol, that is, there is still potential for turnover improvement. This should be included in the discussion.

> As indicated above, the kcat may never reach the kcat observed for vanillyl alcohol. They are different substrates and thus have different chemical properties. A sentence has been added to indicate this (p. 10).

16. 262 and figure 7 – please show the water channels, Which direction is the water accessing, from where?

> We did not mention any channel. Further, there will be no directionality but merely diffusion of water where possible. We indicate in the text and the figure the shielding of the substrate/product intermediate by the introduced E151: it is closer to the alkyl moiety, as can be seen in Figure 7.

17. 262-267 – the description of the effect of structural changes on catalysis is a bit too “naive” to me. Explaining covalent bond formation by molecular crowding seems an oversimplification. The authors should provide simulations supporting such conclusion.

> We use the term ‘crowding’ to indicate that by the Q425S mutation, this part of the substrate binding pocket becomes ‘less crowded’. We do not suggest a ‘molecular crowding’ effect which would involve concentration effects. For clarity we replace ‘crowded’ by ‘packed’ which may better reflect the effect of the mutation. By creating more space, there is room for the alkyl moiety to stay away from the N5, preventing adduct formation. To us, this is a likely explanation. We have considered QM/MM calculation but fear that such analyses will not be conclusive as it will be biased by the imposed mechanism and will at most indicate whether some mechanism is possible, but cannot easily exclude other mechanisms. As a full explanation is not at the heart of this study, we prefer to describe a likely mechanism and effects of the introduced mutations. This is supported by the previous studies (refs #20, 25, 33) on the closely related vanillyl alcohol oxidase for which a similar adduct formation has been observed for a similar substrate (p-cresol). Those data (crystal structure, kinetic analysis) are fully in line with the observations and proposed mechanism in this study.

18. Rather than repeating the results, the discussion section should discuss the obtained results against the landscape of current knowledge. Are there other/better methods of mutant prediction available? How does FRESCO benchmarks against those? Were the difficulties encountered in this study similar to other experience in enzyme repurposing. Are the qualities of the obtained enzyme sufficient for industrial utilization? Does it truly have a chance to “enable biocatalytic valorisation of lignin-derived compounds” (289). Etc. – there are a lot of topics one could focus the discussion at, instead of simply summarizing the results.

> The manuscript has a Conclusion section, in which we try to highlight and put in context the most important results. Parts of the conclusion section has been rewritten. Also, with the reported gram-scale conversions added to the manuscript, we feel that the data convincingly show that the engineered enzyme can become a valuable biocatalyst.

REVIEWER COMMENTS

Reviewer #1 (Remarks to the Author):

The authors revised the manuscript according to the suggestions from the reviewers. Only one question need to be solved.

1. Page 10, line 291: Fig. 9 was not found in the revised manuscript. I think it should be added to the manuscript and numbered as Fig. 8.

Reviewer #2 (Remarks to the Author):

The revisions address many of my comments, but not all of them. Among other considerations, the version of the manuscript I received is missing Figures 8 and 9. Therefore, I was unable to evaluate the added data, particularly that for the additional experiment I had suggested. Comments below are numbered according to my original comments.

Specific comments:

1. In my previous review, I indicated that the English throughout the manuscript needed to be improved. I cited some example without providing an exhaustive list. The authors only addressed the examples. The following additional examples are cited:

“a known flavor and fragrance molecule and flexible precursor” should be “a flavor and fragrance molecule and versatile precursor” (Abstract)

“we set out to engineer eugenol oxidase (EUGO) from *Rhodococcus jostii*” should be “we engineered eugenol oxidase (EUGO) from *Rhodococcus jostii* RHA1” (Abstract)

“can simply use dioxygen as mild oxidant for the catalyzed reactions” should be “it requires only dioxygen as a co-substrate” (p. 4, top)

“duplo” should be “duplicate” (Table 2 footnote)

In the references, species names should be italicized and journal titles should be appropriately abbreviated.

5. I agree that the authors should not evaluate the steady-state kinetic parameters of all 13 “7-fold” variants. However, the D151E/S394V-EUGO5X is a key intermediate in generating PROGO from S394V-EUGO5X. As such, the steady-state kinetic parameters may provide more insight in the generation of PROGO than the single data point in Figure 6 that the authors reference in concluding that D151E/S394V-EUGO5X “showed poor performance”.

8. The authors have performed experiments to establish the biocatalytic utility of PROGO. Unfortunately, Figure 9 was not part of the revised manuscript I received, so I am unable to evaluate the data. In addition, the language describing these experiments could be clearer. For example, it is unclear how *E. coli* cells contain “approximate 24 mg isolated enzyme”. Similarly, the significance of the associated reference (25) is unclear.

Reviewer #3 (Remarks to the Author):

The authors dealt well with the comments of this and the other reviewers which significantly improved the technical part of the manuscript. With the introduced changes the presentation is technically sound and could certainly be of interest for the readership of a more specialized journal. Nevertheless, in my opinion, the communicated data has not enough breadth and influence on the field

to be of interest to a broad readership of Nature Communications. The work soundly solves a a very particular problem of enzyme engineering. Were the work an exemplification of a new conceptual approach to enzyme repurposing it would have merit a broad interest. However, the presented work, even with introduced changes, is just an example of utilization of known methods and approaches. A sound example, but still one of many, and with limited impact in the field. As such, I suggest to reject the manuscript with strong encouragement for publishing in a more specialized journal.

Of minor comments to the revised text:

1. Table 1. Please, change „duplo” to „duplicate”
2. Figure 2. MSA – please, use full name
3. Page 7 (all page numbers reffer to the marked version), MSA – indicate how many sequences / independent sequence clusters were considered
4. Page 8 – change „very high level of conversion” to „80% conversion”
5. Page 10 – Fig. 9 is cited but is not present in the manuscript
6. Page 10 – I do not understand citing reference 25 with regard to PROGO yield. The yield should be determined (or estimated) for the particular experiment performed.
7. Page 10 an 11, section „Preparative scale conversion of 4-n-propylguaiaicol” – provide the amounts of substrate and enzyme in moles (parallel to mg).

Reviewer #1 (Remarks to the Author):

The authors revised the manuscript according to the suggestions from the reviewers. Only one question need to be solved.

1. Page 10, line 291: Fig. 9 was not found in the revised manuscript. I think it should be added to the manuscript and numbered as Fig. 8.

> We have added a figure (Fig. 8 on p. 34) to show the gram-scale conversion that is feasible with the engineered enzyme. We apologize that we had not included it in the previous version.

Reviewer #2 (Remarks to the Author):

The revisions address many of my comments, but not all of them. Among other considerations, the version of the manuscript I received is missing Figures 8 and 9. Therefore, I was unable to evaluate the added data, particularly that for the additional experiment I had suggested. Comments below are numbered according to my original comments.

> We have added a figure (Fig. 8 on p. 34) to show the gram-scale conversion that is feasible with the engineered enzyme. We apologize that we had not included it in the previous version.

Specific comments:

1. In my previous review, I indicated that the English throughout the manuscript needed to be improved. I cited some example without providing an exhaustive list. The authors only addressed the examples. The following additional examples are cited:

“a known flavor and fragrance molecule and flexible precursor” should be “a flavor and fragrance molecule and versatile precursor” (Abstract)

➤ This has been corrected. (p. 2, l. 4-5)

“we set out to engineer eugenol oxidase (EUGO) from *Rhodococcus jostii*” should be “we engineered eugenol oxidase (EUGO) from *Rhodococcus jostii* RHA1” (Abstract)

➤ This has been corrected. (p. 2, l. 7)

“can simply use dioxygen as mild oxidant for the catalyzed reactions” should be “it requires only dioxygen as a co-substrate” (p. 4, top)

➤ This has been corrected. (p. 4, l. 6)

“duplo” should be “duplicate” (Table 2 footnote)

➤ This has been corrected, also for Table 1. (p. 22, l. 3 and p. 23, l. 4)

In the references, species names should be italicized and journal titles should be appropriately abbreviated.

➤ This has been corrected. (all changes can be found in the manuscript in which changes have been marked)

With the help of a native speaker, we have also rewritten and corrected parts of the manuscript to improve the quality of the text. All these minor textual changes can be seen in the track-change version of the submitted manuscript.

5. I agree that the authors should not evaluate the steady-state kinetic parameters of all 13 “7-fold” variants. However, the D151E/S394V-EUGO5X is a key intermediate in generating PROGO from S394V-EUGO5X. As such, the steady-state kinetic parameters may provide more insight in the generation of PROGO than the single data point in Figure 6 that the authors reference in concluding that D151E/S394V-EUGO5X “showed poor performance”.

> We have prepared and determined the steady-state kinetic parameters of the D151E/S394V-EUGO5X mutant. The parameters have been added to Table 2 (p. 23), and are in line with the other observed data. The kinetic parameters are similar to the parent mutant enzyme (S394V-EUGO5X), as it is now mentioned in the text (p. 10, l. 4-5).

8. The authors have performed experiments to establish the biocatalytic utility of PROGO. Unfortunately, Figure 9 was not part of the revised manuscript I received, so I am unable to evaluate the data. In addition, the language describing these experiments could be clearer. For example, it is unclear how E. coli cells contain “approximate 24 mg isolated enzyme”. Similarly, the significance of the associated reference (25) is unclear.

> We have added a relevant figure to illustrate that conversion is nearly complete (Fig. 8). The amount of cells used for the conversion is now indicated in OD600 which makes the previous reference #25 irrelevant (the reference has been removed from this paragraph, it was meant to provide an indication of protein expression level). We have also rewritten parts of the respective paragraph to increase clarity. Changes in text can be seen in the track-change version of the manuscript.

Reviewer #3 (Remarks to the Author):

The authors dealt well with the comments of this and the other reviewers which significantly improved the technical part of the manuscript. With the introduced changes the presentation is technically sound and could certainly be of interest for the readership of a more specialized journal. Nevertheless, in my opinion, the communicated data has not enough breadth and influence on the field to be of interest to a broad readership of Nature Communications. The work soundly solves a very particular problem of enzyme engineering. Were the work an exemplification of a new conceptual approach to enzyme repurposing it would have merit a broad interest. However, the presented work, even with introduced changes, is just an example of utilization of known methods and approaches. A sound example, but still one of many, and with limited impact in the field. As such, I suggest to reject the manuscript with strong encouragement for publishing in a more specialized journal.

Of minor comments to the revised text:

1. Table 1. Please, change „duplo” to „duplicate”

> This has been corrected.

2. Figure 2. MSA – please, use full name

> This has been changed.

3. Page 7 (all page numbers refer to the marked version), MSA – indicate how many sequences / independent sequence clusters were considered

> We have added the information (1001 sequences, 14 clusters) in the Materials and Methods section, and added all sequence identifiers to the Supporting Information.

4. Page 8 – change „very high level of conversion” to „80% conversion”

> This has been changed (p.8, l. 1).

5. Page 10 – Fig. 9 is cited but is not present in the manuscript

> We apologize for this mishap. The correct figure (Fig. 8 on p. 34) has been added.

6. Page 10 – I do not understand citing reference 25 with regard to PROGO yield. The yield should be determined (or estimated) for the particular experiment performed.

> We have now indicated the amount of cells used (OD600 value) on p. 10, l. 33. This information has also been added in the Materials and Methods section (p. 17, l. 26). This makes the previous reference #25 irrelevant (the reference has been removed from this paragraph, it was meant to provide an indication of protein expression level).

7. Page 10 and 11, section „Preparative scale conversion of 4-n-propylguaiacol” – provide the amounts of substrate and enzyme in moles (parallel to mg).

> This has been added.

REVIEWERS' COMMENTS

Reviewer #2 (Remarks to the Author):

The authors have addressed all of my comments. With respect to the previous revision, the additional data are useful and the manuscript reads much better.